# DRIFT: Decompose, Retrieve, Illustrate, then Formalize Theorems

**Meiru Zhang**[*1], **Philipp Borchert**[2], **Milan Gritta**[2], **Gerasimos Lampouras**[2]

[1]Language Technology Lab, University of Cambridge, UK
[2]Huawei Noah's Ark Lab, London, UK
`mz468@cam.ac.uk, philipp.borchert@h-partners.com,`
`milan.gritta@huawei.com, gerasimos.lampouras@huawei.com`

## Abstract

Automating the formalization of mathematical statements for theorem proving remains a major challenge for Large Language Models (LLMs). LLMs struggle to identify and utilize the prerequisite mathematical knowledge and its corresponding formal representation in languages like Lean. Current retrieval-augmented autoformalization methods query external libraries using the informal statement directly, but overlook a fundamental limitation: informal statements lack direct mappings to mathematical theorems and lemmata, nor do those theorems translate trivially into the formal primitives of languages like Lean. To address this, we introduce DRIFT, a novel framework that enables LLMs to decompose informal mathematical statements into smaller, more tractable "sub-components". This facilitates targeted retrieval of premises from mathematical libraries such as Mathlib. Additionally, DRIFT retrieves illustrative theorems to help models use premises more effectively in formalization tasks. We evaluate DRIFT across diverse benchmarks (ProofNet, ConNF, and MiniF2F-test) and find that it consistently improves premise retrieval, nearly doubling the F1 score compared to the DPR baseline on ProofNet. Notably, DRIFT demonstrates strong performance on the out-of-distribution ConNF benchmark, with BEq+@10 improvements of 42.25% and 37.14% using GPT-4.1 and DeepSeek-V3.1, respectively. Our analysis shows that retrieval effectiveness in mathematical autoformalization depends heavily on model-specific knowledge boundaries, highlighting the need for adaptive retrieval strategies aligned with each model's capabilities.

## 1 Introduction

Autoformalization is formulated as a translation task from natural language mathematical descriptions to machine-verifiable statements written in formal languages, such as Rocq (Barras et al., 1997), Isabelle (Paulson, 1994), and Lean (De Moura et al., 2015). Previous work has shown that accurate autoformalization is a critical step towards developing automated theorem proving systems (Lin et al., 2025b; Chen et al., 2025; Xin et al., 2024; Lin et al., 2025c), and ultimately assisting mathematicians in new discoveries (Gouëzel & Shchur, 2019; Leang et al., 2025). Despite recent progress by Large Language Models (LLMs) in informal mathematical reasoning (Ahn et al., 2024; Setlur et al., 2024; Luong & Lockhart, 2025), formal reasoning presents distinct challenges. The strict syntax and necessity for precise alignment between informal concepts and formal definitions mean that even frontier LLMs often fail at autoformalization tasks off-the-shelf (Wu et al., 2025).

Although synthetic data generation could enable the finetuning of LLMs for autoformalization (Jiang et al., 2023; Lin et al., 2025a; Wang et al., 2024; Ying et al., 2024), the knowledge cutoff issue raised by updating formal libraries like Mathlib (Mathlib Community, 2020) makes finetuned models prone to hallucinating about non-existent formal objects that have been renamed, reorganized, or deprecated (Baanen et al., 2025). To bypass these static limitations and support dynamic test-time queries for agentic models, early retrieval-augmented methods addressed this by retrieving similar theorems from external libraries to provide useful syntactic structures and compositional examples.

---

[*]Work conducted during an internship at Huawei Noah's Ark Lab, London.

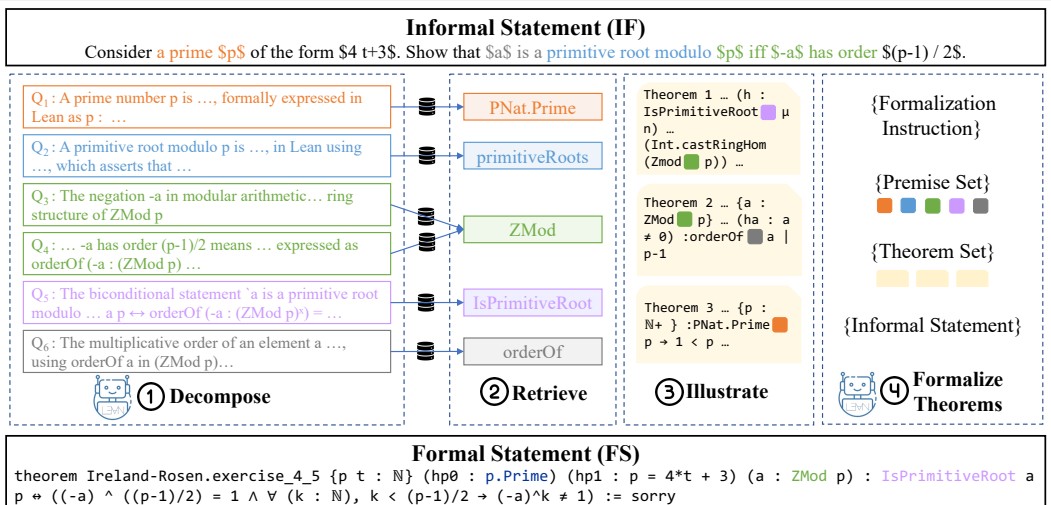

Figure 1: An overview of the DRIFT framework. Given an informal statement, DRIFT operates in four stages: ① **Decompose:** An LLM breaks down the informal statement into atomic, concept-focused sub-queries ($\mathcal{Q}$) (§3.1). ② **Retrieve:** For each sub-query, a dense retriever identifies foundational dependent premises from a formal library (§3.2). ③ **Illustrate:** Greedy selection of a small set of theorems that demonstrate the practical usage of retrieved premises (§3.3). ④ **Formalize Theorems:** Conditioned on all retrieved context, an LLM synthesizes the final formal statement (§3.4).

However, their practical utility as exemplars has been limited by the retrieval methods used to find them. These methods often lack task-specific training data and rely on general-purpose techniques like keyword searching (Agrawal et al., 2022), k-NN (Azerbayev et al., 2023), or pretrained dense retrievers (Zhang et al., 2024). An advance is the introduction of the "**dependency retrieval**" task by Liu et al. (2025) with the RAutoformalizer (RAuto) framework. Similar to the premise selection for proof generation (Yang et al., 2023), this paradigm enables the training of specialized retrievers to identify the exact definitions that formal statements require. However, this new approach created a key trade-off: focusing on individual components meant losing the valuable context provided by full theorem statements. Based on these observations, we identify two main limitations in the current approach to retrieval-augmented autoformalization. First, informal statements are often dense and multifaceted. This **underlying complexity of queries** makes them suboptimal as direct queries for retrieving the precise, atomic definitions required for formalization. Second, even when the correct formal definitions are retrieved, models often **lack the knowledge of contextual usage** required to correctly structure and integrate them into the formal statement.

In information retrieval, query enhancement techniques like query expansion (Chan et al., 2024), pseudo-document generation (Gao et al., 2023), and neural query rewriting (Wang et al., 2025) have demonstrated the effectiveness of reformulating queries to enrich their semantic information. Query decomposition has proven particularly useful for multi-hop question answering as it matches the granularity of the dense query statements with the indexed documents (Ammann et al., 2025). In addition to retrieving correct premises, providing rich context like exemplar proofs can guide proof generation effectively (He et al., 2024; Thakur et al., 2024; Thompson et al., 2025). Despite such advances, these techniques have not yet been systematically applied to dependency retrieval for autoformalization, which still relies on monolithic queries and provides context-free definitions.

We propose DRIFT, a novel framework depicted in Figure 1, that enhances retrieval-augmented autoformalization by adapting query decomposition and context augmentation to address the unique challenges of theorem autoformalization. To tackle the complexity of informal statements, we first decompose statements into a series of simpler, atomic sub-queries. Each sub-query targets a single mathematical concept, transforming a multifaceted retrieval problem into focused, precise searches. To better integrate the dependencies, we contextualize the retrieved definitions with illustrative theorems, giving the model concrete examples of syntax and application patterns.[1]

---

[1] The code and models are available at `https://github.com/Formal-Math-Reasoning/DRIFT` and `https://huggingface.co/Formal-Math-Reasoning/DRIFT-dpr-mathlib`.

**Contributions.** **1)** We introduce DRIFT (**D**ecompose, **R**etrieve, **I**llustrate, then **F**ormalize **T**heorems), a decomposition-driven retrieval-augmented formalization framework that autonomously breaks down informal mathematical statements into atomic sub-queries and contextualizes retrieved premises with demonstrative theorems. This approach bridges the critical gap between formal definition and syntactic usage while transforming monolithic retrieval into a process aligned with the dependency structure of formal mathematics. **2)** Our experiments establish new state-of-the-art in dependency retrieval and autoformalization on ProofNet and ConNF across both frontier LLMs and specialized open-source formalization models, with exceptional performance on the out-of-distribution ConNF benchmark. **3)** Through systematic analysis, we establish that the utility of retrieved dependencies is conditioned on the gap between a model's parametric knowledge and the statement's complexity. These insights reveal critical design considerations for retrieval-augmented systems and point toward the necessity of adaptive strategies that can assess when external knowledge genuinely complements model capabilities.

## 2 RELATED WORK

**Retrieval-augmented Autoformalization.** Early retrieval-augmented autoformalization methods retrieved similar theorems as few-shot examples. For instance, ProofNet (Azerbayev et al., 2023) employs k-NN search, while MS-RAG (Zhang et al., 2024) uses informal-to-informal retrieval with iterative refinement. LTRAG (Hu et al., 2025) retrieves thought-guided theorem pairs for neuro-symbolic formalization. In a key development, Liu et al. (2025) established the "**dependency retrieval**" paradigm, a premise selection task specialized for autoformalization: given an informal statement, retrieve the precise set of formal objects and definitions $\mathcal{P}_{oracle*}$ that are required for its autoformalization from a library $\mathbb{D}$ (e.g., Mathlib). It is essential to distinguish this task from theorem retrieval tools (e.g., LeanSearch (Gao et al., 2024), LeanExplore (Asher, 2025), Moogle (Morph Labs, 2025) or BM25). While some of these systems may also retrieve definitions or structures in practice, they primarily return semantically similar theorems to serve as examples, whereas dependency retrieval explicitly targets the precise, low-level formal definitions (e.g., `IsPrimitiveRoot`, `ZMod`) required for the formal statement. Implementing this paradigm, RAutoformalizer (RAuto) (Liu et al., 2025) demonstrated improvements over non-retrieval methods, yet the evaluation revealed a significant gap compared to oracle systems with ground-truth dependencies. CRAMF (Lu et al., 2025) attempts conceptual mapping between abstraction levels. Critically, however, all existing approaches treat complex statements as monolithic queries, failing to identify distinct mathematical concepts within them.

**Query Decomposition and Enhancement.** Query enhancement has proven effective across retrieval tasks. Query2Doc (Wang et al., 2023) and HyDE (Gao et al., 2023) generate pseudo-documents to expand semantic coverage. LeanSearch (Gao et al., 2024) augments the informal statement by prompting an LLM to translate the query into a detailed statement containing both informal and formal expressions. However, they only evaluated with similar theorem retrieval but not on the downstream formalization. More relevant to our work, query decomposition has shown success in multi-hop question answering (Ammann et al., 2025), where breaking complex questions into sub-queries improves retrieval of distinct information aspects. Zhao et al. (2023) and Jiang et al. (2022) applied similar decompositions to theorem proving, showing the effectiveness of divide-and-conquer in formal math. Despite these successes, no prior work has applied decomposition to informal mathematical statements for autoformalization. Specifically, generic query augmentation methods fail to capture the strict dependency structure within theorems. To bridge this gap, DRIFT employs *atomic decomposition* to directly map mathematical concepts to their formal definitions. Furthermore, unlike similarity-based theorem selection, DRIFT conditions its theorem selection on the retrieved dependencies, ensuring the selected theorems explicitly demonstrate the required syntactic usage of those definitions.

## 3 METHODOLOGY

We introduce DRIFT (**D**ecompose, **R**etrieve, **I**llustrate, then **F**ormalize **T**heorems), a novel four-stage method designed to address the two main limitations of previous retrieval-augmented formalization methods: 1) the complexity of informal statements and 2) the lack of demonstrative examples

for retrieved formal objects. As a first step, an LLM decomposes the informal statement into a set of smaller, granular sub-queries (§3.1). These queries then guide the retrieval of dependent premises from a formal library such as Mathlib (§3.2). In a subsequent, bottom-up illustration step, we find theorems that utilize the retrieved premises, providing in-context demonstrations of their application (§3.3). Finally, the LLM formalizes the original statement, conditioned on all retrieved premises and theorems (§3.4). This process is visualized in Figure 1.

## 3.1 DECOMPOSE

Standard retrieval methods often treat complex informal statements as monolithic queries and simply embed the entire statement. This approach disregards the rich semantic structure within a statement, which may contain multiple distinct mathematical concepts. Their complexity means that a statement could be under-specified, ambiguous, and/or simply too information-dense to be used as is. Compressing the entire statement's meaning into a single dense vector creates a representational bottleneck, risking the loss of nuanced details and focusing the retrieval on only the most salient concepts. We hypothesize that by decomposing an informal statement into its constituent concepts, we can perform a more granular and accurate retrieval for each concept.

To this end, DRIFT begins with a **Decompose** module (panel ① in Figure 1), which is implemented as an LLM tasked with breaking down an informal statement ($IF$) into a set of structured sub-queries, $\mathcal{Q}$, see Equation 1. While the decomposer could be a finetuned model, we prompt off-the-shelf LLMs with few-shot examples in this study.

$$\text{Decomposer}(IF) \rightarrow \mathcal{Q} = \{(q_i, \hat{l}_i)\}_{i=1}^n \qquad (1)$$

where $n$, the number of sub-queries, varies for each informal statement and is determined dynamically by the decomposer. Each sub-query $Q_i = (q_i, \hat{l}_i)$ is designed to isolate a single mathematical concept. As illustrated in Figure 1, the component $q_i$ is a natural language phrase describing the concept (e.g., "A prime number $p$ of the form $4t + 3$ is a prime that leaves remainder 3 when divided by 4") while $\hat{l}_i$ is a predicted formal representation for that concept (e.g., "$p : \mathbb{N}$ with the conditions Nat.Prime $p$ and $p \% 4 = 3$, where $\%$ denotes the modulo operation on natural numbers."). Appending this predicted formal name serves a dual purpose. First, it probes the LLM's parametric knowledge, providing a syntactic "anchor" for the concept. Second, it allows the retriever to jointly leverage the semantics of the natural language phrase $q_i$ and the syntactic cues from $\hat{l}_i$ to identify the correct premise even in the presence of minor inaccuracies in the predicted formal representations.

## 3.2 RETRIEVE

The **Retrieve** module is designed to identify dependent premises from the library that correspond to the concepts isolated in each sub-query. This one-to-one mapping is visualized in the panel ② of Figure 1, which shows each sub-query ($Q_i$) being linked to a formal object (e.g., PNat.Prime). To accomplish this, we implement a dense passage retriever (Karpukhin et al., 2020) using a BGE-M3 (Chen et al., 2024) encoder ($E_\theta$), finetuned on the dependency retrieval task as introduced by Liu et al. (2025). This training objective aligns the informal statements with their formal dependencies, thereby making semantic similarity a strong proxy for logical dependency. We retrieve dependencies by encoding queries and formal library objects into a shared $d$-dimensional embedding space. The vector representations $\mathbf{p} = E_\theta(p)$ for all formal objects $p \in \mathbb{D}$ are pre-computed in an offline step and stored in an efficient search index. At inference, each sub-query is encoded into a vector $\mathbf{q}_i = E_\theta(Q_i)$ and the closest dependent premise $p_i$ is identified by finding the library object that maximizes the cosine similarity, $\phi$, with the query vector. This is formally defined as:

$$p_i = \underset{p \in \mathbb{D}}{\operatorname{argmax}} \left( \phi(\mathbf{q_i}, \mathbf{p}) \right) \qquad (2)$$

where $\phi(\mathbf{q}_i, \mathbf{p}) = \frac{\mathbf{q}_i \cdot \mathbf{p}}{\|\mathbf{q}_i\| \|\mathbf{p}\|}$. The final set of dependent premises, $\mathcal{R}_{\text{DRIFT}}$, is formed by aggregating the top-1 results from each sub-query and removing duplicates: $\mathcal{R}_{\text{DRIFT}} = \bigcup_{i=1}^n \{p_i\}$.

### 3.3 ILLUSTRATE

Retrieving useful formal definitions is a necessary first step; however, it is not sufficient for successful autoformalization. For instance, retrieved definitions like "`def ZMod : ℕ → Type`" indicate information about the concept, but provide no further guidance on its practical application, such as the syntax for declaring a variable of that type (`a : ZMod p`). This gap between definition and usage is a primary source of syntactic and structural errors in LLM-generated statements.

The **Illustrate** module is designed to bridge this gap by providing examples of formal object usage, visualized in Figure 1 (panel ③). Given a premise like `ZMod` from the "Retrieve" step, the module selects illustrative theorems that demonstrate the correct application of `ZMod`, such as "Theorem 2". The module takes the set of retrieved premises $\mathcal{R}_{\text{DRIFT}}$ and a budget $m$ as input, and outputs a small set of illustrative theorems $\mathcal{T}$, where $|\mathcal{T}| \leq m$. The selection process is a greedy algorithm designed to maximize the coverage of the input premises $\mathcal{R}_{\text{DRIFT}}$ as follows. First, we compile a candidate set $\mathcal{T}_{cand}$ of all theorems in the library $\mathbb{D}$ that utilize at least one of the retrieved premises $\mathcal{R}_{\text{DRIFT}}$. In order to ensure relevance and provide a deterministic tie-breaker, we pre-sort this candidate set in descending order of semantic similarity $s(t)$. For each theorem $t \in \mathcal{T}_{cand}$, this similarity score is the cosine similarity between its informal statement $IF_t$ and the original informal statement $IF$, computed using the same encoder $E_\theta$ from the retrieval stage: $s(t) = \phi(E_\theta(IF_t), E_\theta(IF))$.

The final set of illustrative theorems $\mathcal{T}$ is built iteratively. The process begins by initializing an empty set of selected theorems $\mathcal{T}_0 = \emptyset$ and an empty set of covered premises $\mathcal{P}_{cov,0} = \emptyset$. At each step $j = 1, \ldots, m$, we select the theorem $t_j$ that provides the maximal *marginal gain* by including the most new premises in $\mathcal{R}_{\text{DRIFT}}$ not previously covered:

$$t_j = \underset{t \in \mathcal{T}_{cand} \setminus \mathcal{T}_{j-1}}{\operatorname{argmax}} |\mathcal{P}(t) \cap (\mathcal{R}_{\text{DRIFT}} \setminus \mathcal{P}_{cov,j-1})| \tag{3}$$

where $\mathcal{P}(t)$ is the set of premises used in theorem $t$. After selecting $t_j$, the sets are updated for the next iteration: $\mathcal{T}_j = \mathcal{T}_{j-1} \cup \{t_j\}$ and $\mathcal{P}_{cov,j} = \mathcal{P}_{cov,j-1} \cup (\mathcal{P}(t_j) \cap \mathcal{R}_{\text{DRIFT}})$. The process terminates when the budget of $m$ theorems is reached or when no remaining candidate theorem offers additional coverage (i.e., the marginal gain is zero). The final set is defined as $\mathcal{T} = \mathcal{T}_j$ from the last iteration $j$.

### 3.4 FORMALIZE THEOREMS

The final step, **Formalize Theorems** (Figure 1, ④), generates the formal statement $FS$ by conditioning a formalizer on the assembled context $\mathcal{C}$. This module is designed to be flexible and can be implemented with either a finetuned model or a general-purpose LLM. The formalizer compiles information in the following order: $\mathcal{C} = \mathcal{I} \oplus \mathcal{R}_{\text{DRIFT}} \oplus \mathcal{T} \oplus IF$, where $\oplus$ denotes the concatenation operator, $\mathcal{I}$ is a formalization instruction (details in Appendix A.4.2), $\mathcal{R}_{\text{DRIFT}}$ are the retrieved premises, $\mathcal{T}$ are the illustrative theorems and $IF$ is the original informal statement. Conditioned on this prompt, the formalizer then generates the formal statement $FS = \text{Formalizer}(\mathcal{C})$.

## 4 EXPERIMENTAL SETUP

### 4.1 BENCHMARKS

We evaluate DRIFT on three distinct autoformalization benchmarks to test its in-distribution, self-contained, and out-of-distribution performance. While the experiments are conducted in Lean 4, our framework is language-agnostic and adaptable to other formal systems with structured libraries.

**ProofNet (In-Distribution).** We use the ProofNet benchmark (Azerbayev et al., 2023) for in-distribution evaluation. Its 374 theorems, sourced from undergraduate mathematics textbooks, are integrated with the Mathlib library and require an average of 3.39 dependent premises from over 243k formal objects (including 139k theorems). This benchmark tests the framework's primary function, which is to effectively retrieve and utilize dependent premises with demonstration from a large-scale, in-distribution knowledge base.

**MiniF2F (Self-Contained).** We use the MiniF2F-test set (Zheng et al., 2021) to evaluate the framework on self-contained problems. This benchmark consists of 224 Olympiad-style theorems with a

notably *low average* of just 0.43 dependencies from Mathlib.[2] MiniF2F-test serves as a boundary condition, testing the model's core formalization capabilities and its robustness against potentially distracting context when retrieval is not strictly necessary.

**ConNF (Out-of-Distribution).** The OOD challenge refers to evaluation scenarios where neither the retrieval model nor the formalization model has been exposed to the formal objects used in the test data.[3] We therefore evaluate on ConNF, a benchmark curated by Liu et al. (2025) through topological informalization to test OOD generalization. We found no indication of contamination from our zero-shot results on ConNF; however, without access to the underlying training data, this cannot be conclusively validated. This benchmark is based on the `con-nf` library and is *not integrated* with Mathlib.[4] It formalizes a consistency proof for Quine's New Foundations (Quine, 1951), established by Holmes & Wilshaw (2015), and contains 961 research-level theorems requiring an average of 3.92 premises from its distinct library of 1,348 formal objects. ConNF rigorously tests DRIFT's generalization to a novel mathematical domain (Zhang et al., 2024).

## 4.2 BASELINES

We evaluate our proposed framework against three key baselines representing zero-shot, state-of-the-art retrieval, and oracle-level performance. The **no retrieval (zero-shot)** baseline establishes a performance floor. The autoformalization model receives only the informal statement as input, without access to any retrieved context. **DPR (RAuto)** represents the current state-of-the-art. We compare our method to the dependency retrieval module from the RAutoformalizer framework (Liu et al., 2025), which is a finetuned dense passage retriever (DPR) (Karpukhin et al., 2020).[5] The top-5 retrieved premises are provided to the formalizer model as augmented context. The **oracle\* (retrieval)** setting provides an *approximate* upper bound for retrieval. The model is provided with ground-truth dependencies ($\mathcal{P}_{oracle*}$) for each problem, simulating a perfect retriever, as defined by Liu et al. (2025). We mark this setting with an asterisk ($*$) to denote that this oracle is, in fact, imperfect. This is because the provided dependencies are not necessarily optimal or exhaustive for formalization and do not necessarily lead to the best autoformalization performance. As we discuss in Section 5, some of our settings actually outperform this imperfect oracle.

## 4.3 IMPLEMENTATION DETAILS

In this study, we evaluate DRIFT as a lightweight prompting strategy without introducing task-specific biases from finetuning. To comprehensively evaluate our framework, we employ two categories of models: frontier models (primary setup) and specialized models (formalization only). We leverage instruction-following models, such as DeepSeek-V3.1 (DeepSeek-AI, 2024), GPT-4.1 (OpenAI, 2025), and Claude-Opus-4 (Anthropic, 2025a), as both decomposers and formalizers to maintain consistency across the pipeline. These generalist models possess strong in-context learning capabilities and are expected to adapt effectively to the retrieved information provided in the DRIFT prompt. To investigate DRIFT's impact on models with deeper parametric knowledge of formal languages, we also evaluate a specialized open-source model, Goedel-V2-8B (Lin et al., 2025c). Its syntax-heavy finetuning, aimed specifically at producing valid Lean statements (often via an internal Chain-of-Thought mechanism, see Figure 3 in Appendix A.2.10), prevents it from following decomposition instructions. Consequently, we use it strictly as a formalizer, relying on GPT-4.1 to generate the decomposed sub-queries and the resulting retrieved context.

**Decompose.** We construct a few-shot prompt using five expertly verified examples from the Putnam benchmark (Tsoukalas et al., 2024) to instruct LLMs to decompose the informal statements (full details in Appendix A.1.1 and Appendix A.4.1). Each decomposed sub-query consists of a natural language description of a concept and its formal representation predicted by the decomposer with parametric knowledge, as detailed in Section 3.1. We employ a single prompt template across all frontier models to prioritize generalizability. Empirically, this yields consistent improvements

---

[2]We removed 20 examples from the dataset that were either duplicated or failed to compile (Lean v4.18.0).

[3]The models we used, GPT-4.1, Claude-Opus-4, DeepSeek-V3.1, and Goedel-V2-8B, have knowledge cutoff dates of June 2024, March 2025, July 2025, and August 2025, respectively.

[4]The `con-nf` library is available at `https://github.com/leanprover-community/con-nf`.

[5]Model available at `https://huggingface.co/purewhite42/dependency_retriever_f`.

across these diverse models (Table 1 and Table 8), demonstrating framework robustness without the need for model-specific optimization. The number of sub-queries decomposed is not fixed but autonomously decided by the decomposer.

**Retrieve.** The retriever model is a dense passage retriever[6] (DPR) finetuned on Mathlib data to map informal queries to their formal dependencies (Liu et al., 2025). We pre-compute embeddings for all formal declarations in the relevant libraries (Mathlib for ProofNet and MiniF2F-test; `con-nf` for ConNF). This training setup establishes ConNF as an OOD benchmark for the retrieval module. For the DPR baseline, we retrieve the top-5 premises based on the entire informal statement. Unlike fixed-$k$ baselines, the number of premises retrieved by DRIFT is dynamic. By selecting the top-1 candidate for each sub-query and performing deduplication, the final set size is at most $n$, where $n$ is the number of decomposed sub-queries.

**Illustrate.** To demonstrate premise usage in real contexts, we select up to $m = 3$ exemplar theorems using the greedy coverage algorithm described in Section 3.3. This value was selected based on an empirical analysis of diminishing returns in premise coverage, as detailed in Appendix A.2.8.

**Formalize Theorems.** As described in Section 3.4, the formalization prompt combines the original informal statement with the retrieved premises and illustrative theorems. Each premise is presented with its full name, formal declaration, and source code. Each illustrative theorem is included as a pair of its informal and formal statements. This pairing demonstrates both the informal-to-formal alignment and the concrete application of the premises within a theorem instance. To evaluate pass@$k$, we generate 10 formalizations for each problem.

## 4.4 EVALUATION METRICS

We evaluate DRIFT in two stages: a) intrinsically through the performance of dependency retrieval and the selection of illustrative theorems, and b) extrinsically through autoformalization.

**Dependency Retrieval.** The effectiveness of the decomposition is measured by its impact on the dependency retrieval task, as the quality of the decomposed sub-queries directly impacts the relevance of the retrieved premises. We measure the quality of the retrieved premises against the oracle* dependencies using **Precision**, **Recall**, and their harmonic mean, **F1-score**.

**Formalization Correctness.** For formalization, we use **Typecheck (TC)** and **BEq+**. Typecheck measures syntactic correctness, indicating the percentage of the generated statements that are valid and can pass the compiler's type checker (Lu et al., 2024; Azerbayev et al., 2023; Liu et al., 2025). For semantic correctness, we use BEq+ (Poiroux et al., 2025), a symbolic metric that measures the logical equivalence between a predicted formal statement and the ground-truth reference by using deterministic proof tactics to bidirectionally prove that each statement can be transformed into the other. We adopt BEq+ over LLM-based evaluations to avoid stochasticity and ensure compiler-verified reliability. Furthermore, given that large-scale human evaluation is infeasible, BEq+ serves as an effective proxy, demonstrating strong alignment with human judgments (Pearson: 0.974, Kendall: 0.872) (Poiroux et al., 2025). For each metric, we assess performance using pass@1 and pass@10, where pass@$k$ indicates that at least one of $k$ independent generations was successful.

## 5 RESULTS AND DISCUSSION

### 5.1 DEPENDENCY RETRIEVAL

We evaluate the effectiveness of the Decompose and Retrieve modules by looking at their impact on intrinsic performance in dependency retrieval. As detailed in Table 1, we compare DRIFT against a monolithic query baseline that uses the same dense retriever but with the original informal statements as queries. This provides a direct comparison to DRIFT, which retrieves a similar number of premises by taking the union of the top-1 results for each of the 5.21 to 6.42 sub-queries generated by the Decompose module.[7] The results show that decomposition provides a substantial perfor-

---

[6]See Appendix A.1.2 for the training details of the retriever.
[7]The full statistics of the decomposed sub-queries are available at Appendix A.2.7.

| Benchmark | Decomposer | Precision | Recall | F1 |
|---|---|---|---|---|
| ProofNet | - | 11.55 | 17.03 | 13.77 |
| | Claude-Opus-4 | 23.02 | **34.70** | **27.68** |
| | GPT-4.1 | 21.71 | 34.46 | 26.64 |
| | DeepSeek-V3.1 | **24.38** | 30.28 | 27.01 |
| MiniF2F-test | - | 0.36 | 4.12 | 0.66 |
| | Claude-Opus-4 | **2.08** | **23.71** | **3.83** |
| | GPT-4.1 | 1.42 | 15.46 | 2.60 |
| | DeepSeek-V3.1 | 0.98 | 9.28 | 1.78 |
| ConNF | - | 25.12 | 32.06 | 28.17 |
| | Claude-Opus-4 | 30.97 | **41.01** | 35.29 |
| | GPT-4.1 | 31.62 | 40.64 | 35.56 |
| | DeepSeek-V3.1 | **34.67** | 39.39 | **36.88** |

Table 1: Dependency retrieval performances (%) of DRIFT and a no-decomposition baseline ("-"). The baseline queries the retriever directly with the original informal statement. Best results in **bold**.

mance improvement in both precision and recall. Averaged across all decomposer models, DRIFT achieves an absolute improvement of 13.34, 2.08, and 7.74 points over the baseline F1 score on the ProofNet, MiniF2F-test, and ConNF benchmarks, respectively. Regarding the choice of decomposer model, we observe that while Claude-Opus-4 achieves the highest F1 scores on ProofNet (27.68%) and MiniF2F-test (3.83%), the performance variation among the LLMs is marginal. Our findings indicate that frontier LLMs are largely interchangeable for this task, as the top-performing models have a maximum F1 score difference of only 2.05% on any benchmark.

The Illustrate module proves highly effective at selecting a concise set of theorems to demonstrate premise usage. Within a maximum of only three selected theorems ($m = 3$), the algorithm achieves a high average premise coverage rate of 74.59±4.80% across all the decomposers and benchmarks.

## 5.2 FORMALIZATION

For extrinsic performance on autoformalization (see Table 2), DRIFT consistently outperforms both the zero-shot baseline and the strong retrieval-augmented baseline DPR (RAuto) on ProofNet and ConNF benchmarks across all metrics (Typecheck and BEq+, pass@1 and pass@10). Specifically, on ProofNet, GPT-4.1 with DRIFT achieves a BEq+ pass@10 of 21.93%, a 2.74% improvement over DPR (RAuto). This trend holds across all models evaluated, demonstrating that our decomposition-driven approach provides more effective context for the formalization task.

We first examine the behavior of general-purpose frontier models (GPT-4.1 and DeepSeek-V3.1). A key insight is that while frontier models are largely interchangeable as query decomposers, this parity does not extend to the final formalization step. For instance, DeepSeek-V3.1 generally outperforms GPT-4.1 in the zero-shot setting across all benchmarks, suggesting stronger parametric knowledge for direct formalization. However, this trend reverses most significantly on the OOD ConNF benchmark when retrieval is introduced.

On ConNF, all models achieve low zero-shot BEq+ scores (<10%), confirming a severe knowledge gap. Retrieval substantially improves performance, with GPT-4.1 consistently outperforming DeepSeek-V3.1 across all metrics under different retrieval strategies. DRIFT provides a particularly significant improvement, increasing GPT-4.1's BEq+@10 score by 55.57% and even surpassing the oracle* baseline by 3.43%. We hypothesize this is because oracle* provides only necessary premises based on the reference statement, while the illustrative theorems selected with DRIFT provide crucial demonstrations of premise usage. We support this hypothesis with a detailed qualitative analysis of the oracle* baseline's failure modes in Appendix A.3.2.

On the in-distribution ProofNet benchmark, the results are more nuanced. GPT-4.1 surpasses DeepSeek-V3.1 on BEq+@10 when using DRIFT. This pattern suggests that GPT-4.1 is more adept at in-context synthesis, integrating and reasoning over retrieved information to construct formal statements. In contrast to the interchangeability we observed among models as decomposers, the formalization stage reveals clear performance differences. The distinction reinforces that the two pipeline stages rely on distinct LLM capabilities: decomposition leverages natural language reasoning, whereas formalization demands advanced formal reasoning.

| Benchmark | Formalizer | Retrieval | TC@1 | BEq+@1 | TC@10 | BEq+@10 |
|---|---|---|---|---|---|---|
| ProofNet | GPT-4.1 | Oracle* | 58.82 | 20.32 | 79.68 | 27.54 |
| | | Zero-shot | 34.22 | 9.36 | 51.60 | 13.37 |
| | | DPR (RAuto) | 51.60(+17.38) | 14.71(+ 5.35) | 73.53(+21.93) | 19.25(+ 5.88) |
| | | DRIFT | **55.88**(+21.66) | **17.38**(+ 8.02) | **77.01**(+25.41) | **21.93**(+ 8.56) |
| | DeepSeek-V3.1 | Oracle* | 71.12 | 21.93 | 82.09 | 27.54 |
| | | Zero-shot | 60.43 | 15.51 | 71.93 | 20.32 |
| | | DPR (RAuto) | 63.37(+ 2.94) | 17.38(+ 1.87) | 73.53(+ 1.60) | 19.52(− 0.80) |
| | | DRIFT | **72.73**(+12.30) | **18.18**(+ 2.67) | **79.41**(+ 7.48) | **20.59**(+ 0.27) |
| | Goedel-V2-8B | Oracle* | 35.29 | 9.09 | 75.40 | 62.03 |
| | | Zero-shot | 38.50 | **9.09** | 76.74 | 55.08 |
| | | DPR (RAuto) | 35.29(− 3.21) | 6.95(− 2.21) | 78.07(+ 1.33) | 53.48(− 1.60) |
| | | DRIFT | **40.37**(+ 1.87) | 8.56(− 0.53) | **85.03**(+ 8.29) | **59.36**(+ 4.28) |
| MiniF2F-test | GPT-4.1 | Oracle* | 75.45 | 23.66 | 89.29 | 30.36 |
| | | Zero-shot | 69.64 | 23.21 | 84.82 | 28.12 |
| | | DPR (RAuto) | **77.23**(+ 7.59) | **24.55**(+ 1.34) | **92.41**(+ 7.59) | **32.14**(+ 4.02) |
| | | DRIFT | 74.55(+ 4.91) | **24.55**(+ 1.34) | **92.41**(+ 7.59) | 29.02(+ 0.90) |
| | DeepSeek-V3.1 | Oracle* | 77.68 | 23.21 | 87.50 | 28.12 |
| | | Zero-shot | **76.34** | **22.77** | 87.50 | 27.23 |
| | | DPR (RAuto) | 75.89(− 0.45) | **22.77**(± 0.00) | 87.95(+ 0.45) | **27.68**(+ 0.45) |
| | | DRIFT | 74.11(− 2.23) | **22.77**(± 0.00) | **88.84**(+ 1.34) | 24.55(− 2.68) |
| | Goedel-V2-8B | Oracle* | 96.00 | 28.89 | 100.00 | 93.33 |
| | | Zero-shot | 93.33 | 22.22 | **100.00** | **93.33** |
| | | DPR (RAuto) | **95.55**(+ 2.22) | 9.77(−12.45) | **100.00**(+ 0.00) | 26.67(−66.66) |
| | | DRIFT | 93.33(+ 0.00) | **26.67**(+ 4.45) | **100.00**(+ 0.00) | **93.33**(+ 0.00) |
| ConNF | GPT-4.1 | Oracle* | 60.46 | 48.28 | 75.23 | 58.90 |
| | | Zero-shot | 7.28 | 4.47 | 11.45 | 6.76 |
| | | DPR (RAuto) | 24.56(+17.28) | 15.19(+10.72) | 31.95(+20.50) | 20.08(+13.32) |
| | | DRIFT | **65.76**(+58.48) | **54.84**(+50.37) | **77.00**(+65.55) | **62.33**(+55.57) |
| | DeepSeek-V3.1 | Oracle* | 57.34 | 44.22 | 71.28 | 55.15 |
| | | Zero-shot | 13.42 | 8.12 | 17.59 | 11.03 |
| | | DPR (RAuto) | 21.96(+ 8.54) | 12.90(+ 4.78) | 28.20(+10.61) | 17.07(+ 6.04) |
| | | DRIFT | **60.67**(+47.25) | **46.72**(+38.60) | **71.18**(+53.59) | **54.21**(+43.18) |
| | Goedel-V2-8B | Oracle* | 9.99 | 4.37 | 48.80 | 23.10 |
| | | Zero-shot | **16.03** | 2.29 | **71.19** | 10.93 |
| | | DPR (RAuto) | 15.92(− 0.11) | 3.64(+ 1.35) | 64.52(− 6.67) | 16.96(+ 6.03) |
| | | DRIFT | 9.89(− 6.18) | **4.27**(+ 1.98) | 40.89(−30.30) | **19.04**(+ 8.11) |

Table 2: Autoformalization performance on ProofNet, MiniF2F-test, and ConNF. Performance is measured by Typecheck (TC@k) and BEq+@k. We compare DRIFT against zero-shot, DPR (RAuto), and oracle* settings. Colored subscripts indicate improvement (blue) or decrease (red) relative to zero-shot. All values are percentages (%), the best results (excluding the oracle*) are **bold**.

As discussed in Section 4, the MiniF2F-test benchmark presents a distinct profile with an average of only 0.43 library dependencies. This limits the potential for retrieval-based improvements, evidenced by the small gap between the zero-shot and oracle* performance (e.g., a pass@10 gap of 2.24% for GPT-4.1 and 0.89% for DeepSeek-V3.1). Instead, this low-dependency regime reveals the models' high sensitivity to the provided context, which can act as a distractor rather than an aid. We provide a detailed analysis of these failures in Appendix A.3.1, offering a granular error taxonomy that specifically identifies issues like force-fitting and over-complication.

Finally, we evaluate the specialized model, Goedel-V2-8B. On the OOD ConNF benchmark, its zero-shot TC@1 is expectedly low, yet its TC@10 is surprisingly high. However, low BEq+ performance indicates that Goedel-V2-8B relies on its deeper parametric knowledge to produce syntactically valid statements but fails to achieve semantic equivalence. By providing retrieved premises and theorems, DRIFT nearly doubles Goedel-V2-8B's zero-shot BEq+ scores and outperforms DPR (RAuto). This demonstrates that precise retrieval allows even specialized models to focus on the formalization task rather than struggling with parametric retrieval. On in-distribution benchmarks, Goedel-V2-8B achieves strong pass@10 but limited pass@1 accuracy. Furthermore, it is sensitive to distracting context, especially on the low-dependency MiniF2F-test benchmark where its performance degrades severely under DPR (RAuto). In contrast, DRIFT maintains its effectiveness.

| Retrieval | ProofNet | | MiniF2F-test | | ConNF | |
|---|---|---|---|---|---|---|
| | TC@1 | BEq+@1 | TC@1 | BEq+@1 | TC@1 | BEq+@1 |
| **DRIFT (GPT-4.1)** | 55.88 | 17.38 | 74.55 | 24.55 | 65.76 | 54.84 |
| w/o Illustrate | 56.15 (+ 0.27) | 14.17 (- 3.21) | 76.34 (+ 1.79) | 24.55 (± 0.00) | 45.47 (-20.29) | 35.90 (-18.94) |
| w/o Decompose | 50.80 (- 5.08) | 13.64 (- 3.74) | 76.34 (+ 1.79) | 26.34 (+ 1.79) | 59.63 (- 6.13) | 46.72 (- 8.12) |
| w/o Retrieval | 34.22 (-21.66) | 9.36 (- 8.02) | 69.64 (- 4.91) | 23.21 (- 1.34) | 7.28 (-58.48) | 4.47 (-50.37) |
| **DRIFT (DeepSeek-V3.1)** | 72.73 | 18.18 | 74.11 | 22.77 | 60.67 | 46.72 |
| w/o Illustrate | 70.32 (- 2.41) | 15.24 (- 2.94) | 77.68 (+ 3.57) | 21.43 (- 1.34) | 41.31 (-19.36) | 29.34 (-17.38) |
| w/o Decompose | 64.97 (- 7.76) | 15.78 (- 2.40) | 77.68 (+ 3.57) | 20.98 (- 1.79) | 50.36 (-10.31) | 38.81 (- 7.91) |
| w/o Retrieval | 60.43 (-12.30) | 15.51 (- 2.67) | 76.34 (+ 2.23) | 22.77 (± 0.00) | 13.42 (-47.25) | 8.12 (-38.60) |

Table 3: Ablation study of DRIFT using GPT-4.1 and DeepSeek-V3.1. First, we remove the **Illustrate** module (premises retrieved with sub-queries provided in $\mathcal{C}$), then the **Decompose** module (premises retrieved using original informal statement provided in $\mathcal{C}$), and finally all **Retrieval** components. Performance (increase) or (decrease) relative to full models is shown in parentheses.

## 5.3 ABLATION STUDY

In order to isolate and measure the contribution of each component of DRIFT, we conducted a systematic ablation study (Table 3). As expected, removing the illustrative theorems (w/o Illustrate) decreased the BEq+ score on ProofNet and ConNF, which confirms that demonstrations of premise usage are crucial for the formalization correctness beyond just the definitions of the formal objects. Intriguingly, additionally removing the Decompose module (w/o Decompose) does not further degrade performance and even leads to a slight recovery on ConNF and ProofNet in the BEq+ score. We hypothesize that this is because the baseline DPR retrieves a thematically homogeneous (lexically close, though less precise) set of premises via single query retrieval, which may be less distracting than the precise but more diverse set retrieved via decomposition. This reveals a crucial synergy: the illustrative theorems act as a scaffold that helps the model navigate the diverse information retrieved via decomposition. In Appendix A.3.3, we analyze specific instances where using premises retrieved by DRIFT alone failed, whereas the full DRIFT pipeline and the baseline DPR succeeded, further validating the scaffolding hypothesis.

The complex interaction creates a trade-off on the MiniF2F-test benchmark: removing theorems improves syntactic correctness (Typecheck) while degrading logical correctness (BEq+). This further supports the hypothesis that for the simpler, low-dependency problems in MiniF2F-test, adding more context can act as a distractor. Removing external context improves syntactic validity but degrades logical correctness of the generated formal statements. This sensitivity strongly motivates the need for more dynamic and adaptive retrieval strategies. To quantify the theoretical gains of such strategies, we provide an "Oracle Ensemble" analysis in Appendix A.2.9, demonstrating that an adaptive upper bound consistently outperforms individual methods. Future work on agentic frameworks could retrieve select information, judge its utility, and iterate based on compiler feedback.

## 6 CONCLUSION

In this work, we introduced DRIFT, a framework that improves autoformalization by tackling two distinct challenges: the underlying complexity of queries and the lack of contextual usage. Our decomposition-driven retrieval addresses the former by breaking down the informal statement into sub-queries and conducting point-to-point retrieval of its formal dependencies. Concurrently, the Illustrate module resolves the latter by providing illustrative examples to guide the utilization of retrieved premises in theorem instances. This dual approach substantially improves formalization correctness on both complex in-distribution (ProofNet) and out-of-distribution (ConNF) benchmarks, demonstrating its effectiveness as a broadly generalizable and model-agnostic strategy. On a simpler, low-dependency MiniF2F-test benchmark, our method performs comparably to related methods. Our findings suggest future work should focus on dynamic and adaptive retrieval strategies, as well as on agentic frameworks that iteratively refine attempts based on compiler feedback. Grounding such iterative reasoning with robust retrieval is crucial to advancing automated formalization.

## LIMITATIONS

First, our evaluation focuses on the core mechanisms of DRIFT rather than an exhaustive hyperparameter search. We have not extensively investigated the sensitivity of formalization performance to variations in the number of retrieved premises ($k$), the quantity of illustrative theorems ($m$), or the ordering of components within the comprehensive prompt. Second, the decomposer and formalizer were not jointly optimized with the retriever. Due to the scarcity of supervised decomposition data, the decomposer relies on few-shot prompting rather than task-specific finetuning. Furthermore, exploring an end-to-end training paradigm for the entire pipeline remains a direction for future work. Third, our retrieval scope was limited to established formal libraries. While we demonstrated effectiveness on Mathlib (in-distribution) and `con-nf` (OOD), future iterations should integrate a broader range of contexts, including local projects and user-defined codebases, to better support diverse formalization environments. Lastly, the evaluation of autoformalization remains an open challenge. To ensure determinism and reproducibility, we prioritized compiler-verified metrics over LLM-based judges. However, developing more robust metrics that capture stylistic and structural nuances is critical for future research.

## ETHICS STATEMENT

We adhere to the licenses of the data artifacts and models used in this study, as well as to the ICLR code of ethics. Human experts who verified decomposed queries were fairly compensated for their contributions. We acknowledge the risks associated with reasoning LLMs, particularly concerning hallucinations within the decomposition and formalization steps. Errors in decomposition can propagate through the retrieval process, potentially leading to the selection of irrelevant premises or incorrect formalizations. While DRIFT operates within the RAG paradigm to mitigate these risks, it should be viewed as an assistive tool rather than an autonomous source of mathematical truth. Finally, LLMs were used as a writing assistant for improving the language and clarity of this manuscript. The scientific contributions, including all ideas, experiments, and analyses, are the work of the human authors.

## REPRODUCIBILITY STATEMENT

We are committed to the full reproducibility of this study. We release the source code for the DRIFT framework, all curated data artifacts, and our finetuned models under a permissible open-source license. Other key implementation details and hyperparameters are described in Appendix A.1.

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

# A APPENDIX

## A.1 IMPLEMENTATION DETAILS

### A.1.1 FEW-SHOT EXAMPLES FOR DECOMPOSITION

To construct a robust set of few-shot demonstrations for our Decompose module, we strategically selected five problems from the Putnam benchmark (Tsoukalas et al., 2024). These problems were chosen to ensure diversity in both their mathematical domain and the number of underlying premises required for their formal statements.

We decomposed the informal statement of each selected problem into its atomic, logical sub-components using Claude-Opus-4 with extended thinking enabled (Anthropic, 2025b). The decomposition followed a zero-shot prompting strategy guided by a carefully engineered instruction set. Human experts verified each exemplar for correctness and logical atomicity. While the decomposition task lacks a strict ground truth, its effectiveness is empirically validated by significant downstream improvements in premise retrieval (see Table 1, Section 5.1).

This curated set of examples, detailed below, provides the model with varied demonstrations for the decomposition task across number theory, algebra, analysis, and geometry.

- **Number Theory:** putnam_1966_b2
- **Algebra:** putnam_2000_b1
- **Analysis:** putnam_2000_a4, putnam_2015_b1
- **Geometry:** putnam_2003_b5

### A.1.2 RETRIEVER FINETUNING DETAILS

We finetuned the BGE-M3 retriever model (Chen et al., 2024) on the Mathlib 4.7 dataset to specialize it for dependency retrieval in formal mathematics. The finetuning process was executed using the FlagEmbedding library (Chen et al., 2024) on a server equipped with four 32GB GPUs. The complete set of hyperparameters used for this process, including optimizer settings and loss configuration, is provided in Table 4.

**Training Objective and Data Source.** We used the informalized Mathlib dataset provided by RAutoformalizer (Liu et al., 2025), which aligns informal statements with Mathlib 4.7.0 formal declarations. Ground-truth dependent premises were extracted from raw Lean code using **Jixia** (BICMR@PKU AI, 2024), a Lean 4 abstract syntax tree parser. The retriever was trained via standard contrastive loss: informal statements as *anchors*, Jixia-extracted dependencies as *positives* samples, and random library objects as *negatives* samples. Crucially, training was restricted exclusively to Mathlib to guarantee strict isolation from the ConNF dataset.

**Toolchain Versions and Robustness.** We used different Lean versions based on data availability and benchmark constraints. Retriever training used Mathlib 4.7.0 to leverage existing informal-formal pairs. For ProofNet and MiniF2F-test evaluations, we used Lean 4.18.0; the near-100% compilation success rate of gold statements confirms this version disparity introduces no instability. For ConNF, we strictly used Lean 4.7.0 to satisfy its pinned dependencies. Notably, DRIFT avoids the brittleness of rapid Lean updates: unlike finetuned models that memorize static syntax and require costly retraining, DRIFT adapts to new versions via low-cost offline re-indexing of the active library.

### A.1.3 LLM GENERATION PARAMETERS

For all generative tasks, sub-query generation (decomposition) and formal statement generation (formalization), we set the temperature to 0.7 for all the frontier models (GPT-4.1, DeepSeek-V3.1, and Claude-Opus-4) to encourage diverse yet coherent outputs. For the open-sourced Goedel-V2-8B, we set temperature to 0.7, max_token to 16k, and top_p to 0.95. To ensure reproducibility, single-attempt evaluations (pass@1) used a fixed seed of 42. For multi-attempt evaluations (pass@10), we generated ten distinct outputs by using a sequential range of seeds from 42 to 51.

| Category | Hyperparameter | Value |
|---|---|---|
| **Model & Data** | model_name_or_path | bge-m3 |
| | train_data | mathlib 4.7 |
| | query_max_len | 1024 |
| | passage_max_len | 1024 |
| | train_group_size | 4 |
| | sentence_pooling_method | cls |
| **Training** | num_train_epochs | 1 |
| | per_device_train_batch_size | 32 |
| | per_device_eval_batch_size | 4 |
| | learning_rate | $5 \times 10^{-6}$ |
| | warmup_ratio | 0.1 |
| | weight_decay | 0.01 |
| | repetition_penalty | 1.0 |
| | dataloader_drop_last | True |
| | even_batches | True |
| | non_blocking | False |
| | split_batches | False |
| | use_seedable_sampler | True |
| **Loss & Objective** | temperature | 0.02 |
| | normalize_embeddings | True |
| | negatives_cross_device | True |
| | same_benchmark_within_batch | True |
| | unified_finetuning | True |
| | kd_loss_type | m3_kd_loss |
| **Optimizer** | optim | adamw_torch |
| | adafactor | False |
| | adam_beta1 | 0.9 |
| | adam_beta2 | 0.999 |
| | adam_epsilon | $1 \times 10^{-8}$ |

Table 4: Hyperparameters for model fine-tuning.

## A.2 ADDITIONAL RESULTS AND DISCUSSION

### A.2.1 DEPENDENCY RETRIEVAL PERFORMANCE METRICS

We evaluate retrieval performance using standard precision, recall, and their harmonic mean, the F1 score. For a given retrieved set $\mathcal{R}$ and the ground-truth set of oracle* premises $\mathcal{P}_{oracle*}$, precision and recall are defined as: $\text{Precision(P)} = \frac{|\mathcal{P}_{oracle*} \cap \mathcal{R}|}{|\mathcal{R}|}$ and $\text{Recall(R)} = \frac{|\mathcal{P}_{oracle*} \cap \mathcal{R}|}{|\mathcal{P}_{oracle*}|}$ The F1 score provides a single, balanced measure of performance by combining precision and recall: $\text{F1} = 2 \cdot \frac{\text{P} \cdot \text{R}}{\text{P} + \text{R}}$ The composition of the retrieved set $\mathcal{R}$ varies by method.

For baseline retrievers, $\mathcal{R}$ consists of the top-$k$ premises with the highest cosine similarity to the embedding of the full informal statement. For DRIFT, $\mathcal{R}$ is the union of the single best-retrieved premise for each of the $n$ decomposed sub-queries.

### A.2.2 DEPENDENCY RETRIEVAL RESULTS OF RAUTO

This section presents the performance of DPR (RAuto) across both in-distribution (ProofNet, MiniF2F-test) and out-of-distribution (ConNF) benchmarks. The results, detailed in Table 5, highlight a crucial trade-off between specialization and generalization that motivates our proposed approach. When comparing DPR baselines, our retriever without decomposition substantially outperforms DPR (RAuto) on the ConNF benchmark but underperforms on ProofNet and MiniF2F-test. We hypothesize that this discrepancy arises because DPR (RAuto) may be overfitted to Mathlib-specific content.

| Benchmark | Precision | Recall | F1 |
|---|---|---|---|
| ProofNet | 22.89 | 33.75 | 27.28 |
| MiniF2F-test | 0.63 | 7.22 | 1.15 |
| ConNF | 14.01 | 17.88 | 15.71 |

Table 5: Dependency Retrieval performance (%) of DPR (RAuto), the retriever from RAutoformalizer (Liu et al., 2025). Retrieval $k$ is set to 5.

### A.2.3 AUTOFORMALIZATION RESULTS OF CLAUDE-OPUS-4

In the main paper, we evaluated Claude-Opus-4's effectiveness as a decomposer for query decomposition. Here, we provide Claude-Opus-4's formalization results for completeness. Table 6 reports pass@1 performance (TC@1 and BEq+@1) across all benchmarks. Note that pass@10 results for Claude-Opus-4 are not available due to computational costs.

| Benchmark | Retrieval | TC@1 | BEq+@1 |
|---|---|---|---|
| **ProofNet** | Oracle* | 81.28 | 26.20 |
| | Zero-shot | 68.45 | 17.65 |
| | RAuto | 75.67$_{(+7.22)}$ | 19.25$_{(+1.60)}$ |
| | DRIFT | **78.61**$_{(+10.16)}$ | **19.79**$_{(+2.14)}$ |
| **MiniF2F-test** | Oracle* | 93.30 | 35.27 |
| | Zero-shot | 95.09 | 31.25 |
| | RAuto | **95.98**$_{(+0.89)}$ | 30.36$_{(-0.89)}$ |
| | DRIFT | 93.75$_{(-1.34)}$ | **32.59**$_{(+1.34)}$ |
| **ConNF** | Oracle* | 62.75 | 50.36 |
| | Zero-shot | 13.32 | 8.53 |
| | RAuto | 26.64$_{(+13.32)}$ | 18.52$_{(+9.99)}$ |
| | DRIFT | **72.32**$_{(+59.00)}$ | **60.35**$_{(+51.82)}$ |

Table 6: Autoformalization performance on ProofNet, MiniF2F-test, and ConNF. Performance is measured by Typecheck (TC@1) and BEq+@1. We compare DRIFT against zero-shot, DPR (RAuto), and oracle* settings. Colored subscripts indicate improvement (blue) or decrease (red) relative to zero-shot. All values are percentages (%), the best results (excluding the oracle*) are **bold**. The pass@10 experiments for Claude were omitted due to funding constraints.

### A.2.4 THE ROLE OF ILLUSTRATIVE THEOREMS AS A SCAFFOLD

As presented in Table 3, removing the Decompose module (w/o Decompose: reverting to the baseline DPR) does not degrade performance further. In fact, on ConNF and ProofNet, it leads to slight recovery in the BEq+ scores compared to the "w/o Illustrate" setting. We hypothesize this is due to the nature of retrieval noise. The baseline DPR, using a single query, retrieves a thematically clustered set of premises. While its precision is lower, its noise is homogeneous and may be less distracting to the LLM. Our decomposition method retrieves a more diverse set of premises. While this captures more correct dependencies (higher recall and precision), the accompanying noise is also more varied. This reveals a crucial synergy: the selected theorems in the Illustrate module act as a contextual scaffold, helping the model navigate the diverse information retrieved by the decomposer. Without this guidance, the varied noise can outweigh the benefit of improved retrieval.

### A.2.5 SCALING PERFORMANCE WITH SAMPLING

Across all experiments in Table 2, we observe a consistent and significant gap between pass@1 and pass@10 results. For instance, performance on ProofNet improved by an average of 27.20% across all settings and formalizer models. This large uplift underscores the potential for enhancing performance through sampling-based methods at test time. This suggests that performance could be

further scaled by integrating our method into an agentic framework equipped with a verifier (e.g., Typecheck correctness).

Notably, DeepSeek-V3.1's performance of pass@10 saturates more quickly than Claude-Opus-4's on the ProofNet benchmark. Its zero-shot pass@10 score is only 0.27% lower than the DRIFT score. This suggests that with sufficient sampling, the model can sometimes recover the necessary knowledge parametrically. We anticipate a similar, albeit slower, trend for the larger Claude-Opus-4 and GPT-4.1 models if the number of attempts were increased further.

### A.2.6   PARAMETRIC RETRIEVAL

| Retrieval | ProofNet | | MiniF2F-test | | ConNF | |
|---|---|---|---|---|---|---|
| | TC@1 | BEq+@1 | TC@1 | BEq+@1 | TC@1 | BEq+@1 |
| DRIFT (GPT-4.1) | 55.88 | 17.38 | 74.55 | 24.55 | 65.76 | 54.84 |
| Parametric Retrieval | 43.85 (-12.03) | 13.64 (- 3.74) | 63.84 (-10.71) | 20.09 (- 4.46) | 10.41 (-55.35) | 3.64 (-51.20) |
| w/o Retrieval | 34.22 (-21.66) | 9.36 (- 8.02) | 69.64 (- 4.91) | 23.21 (- 1.34) | 7.28 (-58.48) | 4.47 (-50.37) |
| DRIFT (DeepSeek-V3.1) | 72.73 | 18.18 | 74.11 | 22.77 | 60.67 | 46.72 |
| Parametric Retrieval | 69.25 (- 3.48) | 19.79 (+ 1.61) | 76.79 (+ 2.68) | 24.55 (+ 1.78) | 10.51 (-50.16) | 5.93 (-40.79) |
| w/o Retrieval | 60.43 (-12.30) | 15.51 (- 2.67) | 76.34 (+ 2.23) | 22.77 ($\pm$ 0.00) | 13.42 (-47.25) | 8.12 (-38.60) |

Table 7: Performance comparison of the full DRIFT model, parametric retrieval baseline, and zero-shot using GPT-4.1 and DeepSeek-V3.1 with pass@1. Values in parentheses show performance change relative to the full DRIFT model.

To disentangle the contributions of an LLM's internal (parametric) knowledge and external (retrieved) knowledge, we conducted a "parametric retrieval" experiment. In this setting, we prompted the formalizer models only with the decomposed sub-queries, omitting the retrieved premises and illustrative theorems. This setup probes whether the structured sub-queries alone are sufficient to guide the models to access their own latent knowledge for the formalization task.

The results in Table 7 indicate that external knowledge from retrieval remains largely indispensable and cannot be fully substituted by the LLM's internal knowledge alone. However, we observe a notable distinction between the models. DeepSeek-V3.1 demonstrates a stronger grasp of the required formal knowledge; for this model, the sub-queries appear to function as a Chain-of-Thought-style prompt, structuring its reasoning process and thereby improving formalization accuracy. This aligns with our earlier finding that DeepSeek-V3.1's zero-shot performance with sufficient sampling (pass@10) approaches its retrieval-augmented performance, suggesting it often possesses the necessary formal knowledge but requires effective prompting to surface it.

Crucially, this ablation counters the data contamination hypothesis, which posits that retrieval merely primes models to recall memorized solutions on in-distribution benchmarks. Under this hypothesis, explicit sub-queries alone should suffice to trigger correct recall. However, the significant performance gap between Parametric Retrieval and full DRIFT (e.g., GPT-4.1 improving from 13.64% to 17.38% on ProofNet) confirms that retrieved context provides essential syntactic and semantic information absent from the model's parameters, rather than simply acting as a memory trigger.

### A.2.7   STATISTICS OF DECOMPOSED SUB-QUERIES

To better understand the Decompose module and its behavior, we analyzed the number of sub-queries generated when decomposing informal statements from our three benchmarks: ProofNet, MiniF2F-test, and ConNF with different LLMs as decomposers.

The results, summarized in Table 8, show that different models produce a varying number of sub-queries. GPT-4.1 tends to generate the most detailed decompositions, with an average of 6.27 sub-queries, while DeepSeek-V3.1 produces the most concise ones, averaging 5.06. Furthermore, the complexity of the benchmark appears to influence the decomposition length. Statements from the ConNF benchmark, which covers frontier mathematical research, consistently required more sub-queries (6.42 on average) across all models, likely reflecting their greater conceptual density compared to the undergraduate-level problems in ProofNet (5.69) and the more self-contained problems in MiniF2F-test (5.21).

| Model | ProofNet | MiniF2F-test | ConNF | Model Avg. |
|---|---|---|---|---|
| Claude-Opus-4 | 5.84 | 5.59 | 6.52 | 5.98 |
| GPT-4.1 | 6.39 | 5.40 | 7.03 | 6.27 |
| DeepSeek-V3.1 | 4.83 | 4.65 | 5.71 | 5.06 |
| Benchmark Avg. | 5.69 | 5.21 | 6.42 | 5.77 |

Table 8: Average number of decomposed sub-queries generated by different LLMs as decomposers across three benchmarks. The final row and column show the average values for each benchmark and model, respectively.

### A.2.8 DIMINISHING RETURNS OF INCREASING ILLUSTRATIVE THEOREMS

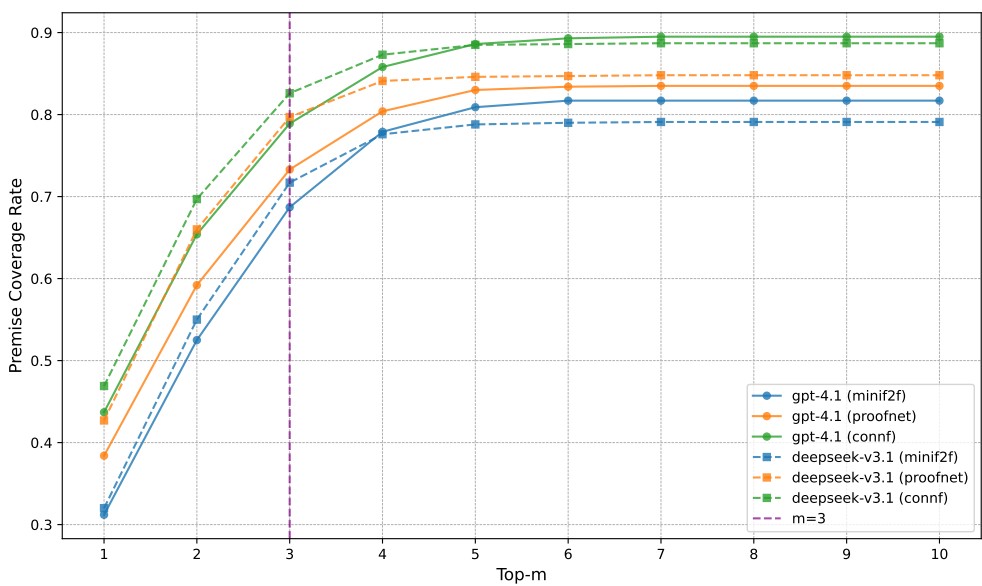

Figure 2: Premise Coverage Rate vs. Top-$m$. The dashed purple line marks the selected budget of $m = 3$, indicating the point of diminishing returns.

We set the illustration budget to $m = 3$ to balance premise coverage against contextual noise. To validate this, we empirically analyzed the premise coverage rate across varying budgets $m$. As illustrated in Figure 2, $m = 3$ marks the "elbow" of the curve, representing a critical point of diminishing returns. Although coverage increases marginally up to $m = 5$, the curve flattens significantly after $m = 3$, indicating that the first three theorems capture most retrieved premises. Furthermore, our ablation study (Table 3, Section 5.3) supports this threshold, demonstrating that excessive context degrades performance in low-dependency regimes (e.g., MiniF2F-test).

### A.2.9 THEORETICAL GAINS OF ADAPTIVE RETRIEVAL ON SELF-CONTAINED BENCHMARKS

To quantify the theoretical gains of adaptive retrieval, a system dynamically choosing whether to skip retrieval for self-contained problems or employ DRIFT for dependency-heavy ones, we evaluated an "Oracle Ensemble" on MiniF2F-test. This metric represents the performance upper bound of a perfect classifier selecting the optimal strategy (Zero-Shot or DRIFT) for each instance. As shown in Table 9, the Oracle Ensemble consistently outperforms both the Zero-Shot baseline and DRIFT. For example, using Claude-Opus-4, the ensemble improves the BEq+@1 score from 32.59% (DRIFT) to 35.27%. This confirms the approaches are complementary: DRIFT provides crucial scaffolding for complex formalization, while Zero-Shot avoids introducing spurious dependencies in self-contained problems. Although training an adaptive classifier (e.g., Self-RAG (Asai et al., 2024)) is beyond the

| Model | Metric | Zero-Shot | DRIFT | Oracle Ensemble (Adaptive Upper Bound) |
|---|---|---|---|---|
| Claude-Opus-4 | TC@1 | 95.09 | 93.75 | **95.98** |
| | BEq+@1 | 31.25 | 32.59 | **35.27** |
| GPT-4.1 | TC@1 | 69.64 | 74.55 | **81.70** |
| | BEq+@1 | 23.21 | 24.55 | **25.89** |

Table 9: "Oracle Ensemble" Performance on MiniF2F-test (Pass@1). The Oracle Ensemble represents the best-case scenario for adaptive retrieval, selecting the best result between Zero-Shot and DRIFT per problem. Best results are **bold**.

scope of this work, these findings establish a strong theoretical motivation for future research into dynamic retrieval gating.

### A.2.10 ANALYSIS OF GOEDEL-V2-8B'S INTERNAL REASONING

As discussed in the Section 4, Goedel-V2-8B's specialized finetuning makes it highly reliant on parametric knowledge. We observe that it achieves its strong zero-shot typecheck performance via an implicit Chain-of-Thought "thinking" mechanism, an example of which is illustrated in Figure 3.

While this internal reasoning already includes decomposition and self-reflection, it lacks the precision of DRIFT's explicit retrieval. For instance, on the ProofNet benchmark, Goedel-V2-8B fails to effectively leverage external oracle* context (ProofNet BEq+@1 remains 9.09%), relying instead on extensive sampling (pass@10) to generate correct solutions.

As noted in Section 5.2, this internal "thinking" mechanism, while syntactically robust, frequently hallucinates dependencies when pushed out-of-distribution to novel domains like ConNF. In contrast, providing external knowledge via DRIFT nearly doubles Goedel-V2-8B's single-pass precision on ConNF (BEq+@1 improves from 2.29% to 4.27%, BEq+@10 improves from 10.93% to 19.04%). This further highlights the necessity of the explicit, non-distracting context provided by DRIFT to bridge the gap between syntactic validity and semantic correctness.

### A.3 QUALITATIVE ANALYSIS

### A.3.1 MINIF2F-TEST FAILURE ANALYSIS

To investigate when retrieval harms MiniF2F-test performance, we conducted an error analysis. Across 672 examples from all benchmarked models, we identified 23 instances where the zero-shot baseline succeeded but DRIFT failed. Averaging just 22.20 words and requiring no explicit quantifiers, these informal statements exhibit very low complexity. Notably, in 12 of these 23 cases, DRIFT generated statements that passed typecheck but failed BEq+.

This analysis revealed three primary failure modes on MiniF2F-test. First, *over-complication* from retrieval noise can confuse the model on straightforward problems, whereas the zero-shot baseline yields cleaner statements. Second, *variable naming issues* arise when retrieved context introduces inconsistent identifiers. Finally, *formalization style mismatches* occur when the model adopts structural patterns from retrieved theorems that deviate from the reference, causing BEq+ to miss semantic equivalences. Selected examples illustrating these failures are provided below.

**Example 1: mathd_algebra_432 (Claude-Opus-4)**

The Problem: Expand $(x + 3)(2x - 6)$

Zero-shot (correct):

```
theorem mathd_algebra_432 (x : ℝ) : (x + 3) * (2 * x - 6) = 2 * x^2 - 18
```

DRIFT (failed typecheck).

```
theorem mathd_algebra_432 : (X + 3) * (2 * X - 6) = 2 * X^2 - 18
```

---

**Analysis of Goedel-V2-8B's "Think" Capability**

**Input (Informal Statement):**
If $r$ is rational ($r \neq 0$) and $x$ is irrational, prove that $rx$ is irrational.

---

**Model Internal Reasoning (`<think>`):**

Let me analyze this problem step by step. The problem is asking to prove that if $r$ is rational (and nonzero) and $x$ is irrational, then $rx$ is irrational.

**In Lean 4, I need to:**

1. Define what it means for a number to be rational
2. Define what it means for a number to be irrational
3. State the theorem that if $r$ is rational (nonzero) and $x$ is irrational, then $rx$ is irrational

**In Lean 4:**

- Rational numbers are represented by $\mathbb{Q}$
- The condition "$r$ is rational and nonzero" can be written as `r : $\mathbb{Q}$` and `r ≠ 0`
- Irrational numbers are represented by the complement of rationals, or we can use `Irrational x` which means $x \notin \mathbb{Q}$ (or equivalently `¬(x : $\mathbb{Q}$)`) when $x$ is a real number

*[Reasoning about Types and Coercion]*
Let me think about the types. Since we're dealing with irrational numbers, we're working in the real numbers $\mathbb{R}$. The rational number $r$ can be coerced into $\mathbb{R}$, and $x$ is a real number.

*[Self-Correction]*
Wait, let me double-check the definition of irrational in Lean 4. I believe `Irrational` is defined as `¬(x : $\mathbb{Q}$)` when $x$ is real... Actually, let me be more careful about the coercion. When we have `r : $\mathbb{Q}$` and `x : $\mathbb{R}$`, the expression `r * x` should work because there's a coercion from $\mathbb{Q}$ to $\mathbb{R}$.

---

**Generated Formal Statement:**
```
theorem : forall (r : $\mathbb{Q}$) (x : ℝ), r ≠ 0 ->
    Irrational x -> Irrational (↑r * x) := by sorry
```

Figure 3: An Example of the Goedel-V2-8B's internal reasoning. The model first outlines the necessary definitions in natural language, then maps them to specific Lean 4 syntax, and finally reasons about type coercion (Real vs. Rational) before generating the compilable code.

The statement generated by DRIFT failed because of the undefined variable "X"

---

**Example 2: mathd_algebra_143 (Claude-Opus-4)**

The Problem: If $f(x) = x + 1$ and $g(x) = x^2 + 3$, what is $f(g(2))$?

Zero-shot (correct):

```
theorem mathd_algebra_143 (f g : ℝ → ℝ) (hf : ∀ x, f x = x + 1) (hg : ∀
    x, g x = x^2 + 3) : f (g 2) = 8
```

DRIFT (failed BEq+; false negative of BEq+):

```
theorem mathd_algebra_143 :
    let f : ℝ → ℝ := fun x => x + 1
    let g : ℝ → ℝ := fun x => x^2 + 3
    f (g 2) = 8
```

Both statements are semantically equivalent to the ground-truth, BEq+ fails to classify DRIFT's statement correctly.

**Example 3: amc12_2000_20 (DeepSeek-V3.1)**

The problem: System of equations with $xyz$

Zero-shot (correct):

```
theorem amc12_2000_20 (x y z : ℝ) (hx : x > 0) (hy : y > 0) (hz : z > 0)
    (h1 : x + 1/y = 4) (h2 : y + 1/z = 1) (h3 : z + 1/x = 7/3) : x * y *
    z = 1
```

DRIFT (failed typecheck):

```
theorem amc12_2000_20 : ∃! (x y z : ℝ), x > 0 ∧ y > 0 ∧ z > 0 ∧ x + 1/y =
    4 ∧ y + 1/z = 1 ∧ z + 1/x = 7/3 ∧ x * y * z = 1
```

### A.3.2 DRIFT OUTPERFORMING ORACLE* RETRIEVAL

As shown in Table 2, DRIFT can surpass the Oracle* baseline on ConNF. We hypothesize that ground-truth premises alone are insufficient because they lack necessary usage context. DRIFT addresses this by providing retrieved theorems that serve as syntactic scaffolding.

Across our evaluated models, DRIFT succeeds where Oracle* fails in 205, 171, and 184 out of 961 cases, respectively. On average, Oracle* failures consist of typecheck errors ($\sim$78%, often missing namespaces or types) and semantic equivalence issues ($\sim$22%, failing BEq+ despite typechecking). Notably, DRIFT generates longer statements that correctly incorporate namespace qualifications, type class instances, proper type coercions, and explicit type annotations, thereby significantly reducing formalization errors.

The cases below illustrate that ground-truth premises cannot guarantee correct formalization when success relies on subtle usage patterns absent from dependency names. DRIFT overcomes this limitation by leveraging retrieved theorems as structural scaffolding. These examples explicitly demonstrate correct syntax (e.g., namespace usage, type coercions like $\uparrow \beta$) and proper argument binding (implicit, explicit, or typeclass). Furthermore, they reveal hidden contextual requirements, such as non-obvious typeclass instances (e.g., `[ConNF.FOAAssumptions]`), and guide accurate semantic formulations not explicitly detailed in the informal statement (e.g., distinguishing set versus type cardinality).

**Example 1: `ConNF.Pretangle.ofCoe_inj` (Claude-Opus-4)**

The Problem: `ConNF.Pretangle.ofCoe_inj` is a theorem in Lean 4 that states the injectivity of the `ConNF.Pretangle.ofCoe` function in the context of Constructive Ordinal Notation (ConNF). This theorem asserts that two pretangles a and b are equal if and only if their images under the `ConNF.Pretangle.ofCoe` function are equal.

Gold Statement:

```
@[simp]theorem ConNF.Pretangle.ofCoe_inj [ConNF.Params ] {α : ConNF.Λ}
    {a : ConNF.Pretangle ↑α} {b : ConNF.Pretangle ↑α
    }:ConNF.Pretangle.ofCoe a = ConNF.Pretangle.ofCoe b ↔ a = b := by
    sorry
```

Oracle* Dependent Premises:

```
ConNF.Params
ConNF.Pretangle
```

Oracle* (failed typecheck): The failure here is a type signature mismatch. The generated statement defined the universal level $\alpha$ as a generic `TypeIndex`, whereas the library requires the specific model index type (ConNF.$\Lambda$). Consequently, the model failed to apply the necessary type coercion ($\uparrow \alpha$) required for `Pretangle` in this context. The premise list provided the names and declaration of those premises, but not the specific type constraints to apply.

```
theorem Pretangle.ofCoe_inj [Params] {α : TypeIndex} {a b : Pretangle α}
    : Pretangle.ofCoe a = Pretangle.ofCoe b ↔ a = b := by sorry
```

DRIFT (correct):

```
theorem ConNF.Pretangle.ofCoe_inj [ConNF.Params] {α : ConNF.Λ} {a b :
    ConNF.Pretangle α} : ConNF.Pretangle.ofCoe a = ConNF.Pretangle.ofCoe
    b ↔ a = b := by sorry
```

The key retrieved theorems: The retrieved theorem `ConNF.Pretangle.toCoe_inj` provided a structural template.

```
theorem toCoe_inj {a b} : (toCoe a : Pretangle α) = toCoe b ↔ a = b :=
    toCoe.injective.eq_iff
```

---

### Example 2: `ConNF.StructAction.refine_precise` (DeepSeek-V3.1)

The Problem: The theorem `ConNF.StructAction.refine_precise` states that if $\varphi$ is a $\beta$-structural action that satisfies the lawfulness condition for each $\beta$-extended index, then the refined $\beta$-structural action `ConNF.StructAction.refine` $\varphi$ h$\varphi$ is precise, meaning it assigns a precise near-litter action to each $\beta$-extended index.

Gold Statement:

```
theorem ConNF.StructAction.refine_precise [ConNF.Params ] {β :
    ConNF.TypeIndex} {φ : ConNF.StructAction β} {hφ :
    ConNF.StructAction.Lawful φ}:ConNF.StructAction.Precise
    (ConNF.StructAction.refine φ hφ) := by sorry
```

Oracle* Dependent Premises:

```
ConNF.Params
ConNF.StructAction.refine
ConNF.StructAction
ConNF.StructAction.Precise
ConNF.StructAction.Lawful
```

Oracle* (failed BEq+): The model failed the bidirectional equivalence check due to argument structure. It incorrectly typed the parameter $\beta$ as `ConNF.Λ` instead of `ConNF.TypeIndex` and failed to structure the hypothesis $h\varphi$ as an instance-implicit argument.

```
theorem ConNF.StructAction.refine_precise [ConNF.Params] {β : ConNF.Λ}
    (φ : ConNF.StructAction β) (hφ : ConNF.StructAction.Lawful φ) :
    ConNF.StructAction.Precise (ConNF.StructAction.refine φ hφ) := by
    sorry
```

DRIFT (correct):

```
theorem ConNF.StructAction.refine_precise [ConNF.Params] {β :
    ConNF.TypeIndex} {φ : ConNF.StructAction β} {hφ :
    ConNF.StructAction.Lawful φ} : ConNF.StructAction.Precise
    (ConNF.StructAction.refine φ) := by sorry
```

The key retrieved theorems: The retrieved theorem `ConNF.NearLitterAction.refine_precise` showed the correct usage pattern for "precise" predicates, implicitly suggesting the argument structure that allowed DRIFT to match the library's conventions.

```
theorem refine_precise : Precise (φ.refine hφ) := fillAtomOrbits_precise
    _ (fillAtomRange_symmDiff_subset_ran hφ)
```

---

### Example 3: `ConNF.CodingFunction.mem_of_smul_mem` (GPT-4.1)

The Problem: In the context of Constructive Ordinal Notation (ConNF), the theorem `ConNF.CodingFunction.mem_of_smul_mem` asserts that for any `CodingFunction` $\chi$ at level $\beta$, if $\rho$ · S (where $\rho$ is an `Allowable` and S is a `Support`) is in the domain of $\chi$, then S itself must also be in the domain of $\chi$.

Gold Statement:

```
theorem ConNF.CodingFunction.mem_of_smul_mem [ConNF.Params ]
    [ConNF.Level] [ConNF.FOAAssumptions] {β : ConNF.Λ} [ConNF.LeLevel ↑β
    ] {χ : ConNF.CodingFunction β} {S : ConNF.Support ↑β} {ρ :
    ConNF.Allowable ↑β} (h : ρ · S ∈ χ) :S ∈ χ := by sorry
```

Oracle* Dependent Premises:

```
ConNF.Params
ConNF.LeLevel
ConNF.CodingFunction
ConNF.Support
```

Oracle* (failed typecheck): The model failed to compile the statement because it missed critical, non-obvious constraints: the global typeclass instance [ConNF.FOAAssumptions] and the specific type coercion from $β$ to $↑β$ for the "LeLevel" instance.

```
theorem ConNF.CodingFunction.mem_of_smul_mem [ConNF.Params]
    [ConNF.Level] {β : ConNF.Λ} [ConNF.LeLevel β] (χ :
    ConNF.CodingFunction β) (ρ : ConNF.Allowable β) (S : ConNF.Support β
    ) : (χ.decode (ρ · S)).isSome → (χ.decode S).isSome := by sorry
```

DRIFT (correct):

```
theorem ConNF.CodingFunction.mem_of_smul_mem [ConNF.Params]
    [ConNF.Level] [ConNF.FOAAssumptions] {β : ConNF.Λ} [ConNF.LeLevel ↑β
    ] (χ : ConNF.CodingFunction β) (ρ : ConNF.Allowable ↑β) (S :
    ConNF.Support β) : S ∈ χ.domain → ρ · S ∈ χ.domain := by sorry
```

The key retrieved theorems: The retrieved theorem `ConNF.CodingFunction.supports_decode` demonstrated the interaction between `CodingFunction` $β$, `Support` $β$, and `Allowable` $β$.

```
theorem supports_decode {χ : CodingFunction β} (S : Support β) (hS : S ∈
    χ) : Supports (Allowable β) (S : Set (Address β)) ((χ.decode S).get
    hS) := χ.supports_decode' S hS
```

While not identical, it provided a contextual example of how these types work together, guiding the model to infer the correct, more complex signature.

---

### Example 4: `ConNF.mk_nearLitter''` (Claude-Opus-4)

The problem: The size of each near-litter in the context of Constructive Ordinal Notation (ConNF) is equal to the cardinality of the type $κ$.

Gold Statement:

```
@[simp]theorem ConNF.mk_nearLitter'' [ConNF.Params ] (N :
    ConNF.NearLitter) :Cardinal.mk ↑N = Cardinal.mk ConNF.κ := by sorry
```

Oracle* Dependent Premises:

```
ConNF.Params
Cardinal.mk
```

Oracle* (failed BEq+): The Oracle* based statement committed a semantic formulation error. It hallucinated a theorem equating the cardinality of the *Type* `ConNF.NearLitter` to $κ$. The problem asks about the size of individual near-litter sets (coerced from "N"), not the cardinality of the type containing all near-litters.

```
theorem ConNF.mk_nearLitter'' [ConNF.Params] : Cardinal.mk
    ConNF.NearLitter = Cardinal.mk ConNF.κ := by sorry
```

DRIFT (correct):

```
theorem ConNF.mk_nearLitter'' [ConNF.Params] (L : ConNF.Litter) (s : Set
    ConNF.Atom) (h : ConNF.IsNearLitter L s) : Cardinal.mk s =
    Cardinal.mk ConNF.κ := by sorry
```

**The key retrieved theorems:** DRIFT retrieved `ConNF.mk_litterSet`, which shows that cardinality theorems in this library typically apply to a specific set $s$ (e.g., "litterSet L") rather than types. This scaffolding helped the model correctly formulate the theorem using an explicit set $s$ and the witness hypothesis `ConNF.IsNearLitter L s`.

```
theorem mk_litterSet (L : Litter) : #(litterSet L) = #κ := Cardinal.eq.2
    ⟨litterSetEquiv L⟩
```

### A.3.3 FAILURE ANALYSIS FOR ABLATION STUDY (CONNF)

Our ablation study (Section 5.3, Table 3) shows that on ConNF and MiniF2F-test, removing the Decompose module after removing Illustrate does not further degrade performance. As noted in Appendix A.3.1, MiniF2F-test relies minimally on dependent premises, making additional context a distraction and performance fluctuations largely stochastic. To understand this phenomenon deeply, we analyze ConNF below, examining cases where both the "Base Retriever" (DRIFT w/o Illustrate and w/o Decompose) and "Full DRIFT" succeed, but "DRIFT Premises Only" (w/o Illustrate) fails. Despite DRIFT achieving higher global recall (40.6% vs. 32.0% for GPT-4.1), this breakdown highlights qualitative differences: the Base Retriever typically finds "lexically close" helper lemmas (e.g., `invFun_as_coe`), whereas DRIFT retrieves a more diverse set of premises across namespaces. Consequently, DRIFT Premises Only fails due to either (1) local recall gaps (missing specific helpers found by Base) or (2) application gaps (retrieving broader definitions but lacking usage examples). Full DRIFT bridges these gaps via theorem scaffolding.

These comparisons reveal that even when DRIFT Premises Only retrieves the necessary definitions (Examples 1 and 3), it suffers from application gaps, failing to apply them correctly. The Base Retriever sometimes avoids this by finding exact-match lemmas (Example 2) or benefiting from favorable ranking (Example 1). However, Full DRIFT remains the most robust; its retrieved theorems provide crucial application knowledge, such as templates, style guides, and usage examples, enabling the model to synthesize correct solutions even without exact-match lemmas or when prone to syntax hallucinations.

A key driver of these differences is premise diversity. The Base Retriever reliably locates lexically close helper lemmas within exact or adjacent namespaces (e.g., `PartialPerm.invFun_as_coe` or `Cardinal.*`). Conversely, DRIFT retrieves a broader set of premises, encompassing wider concepts (e.g., `Set`, `PartialOrder`, `PFun.image`) and distinct namespaces (e.g., `Equiv` vs. `PartialPerm`). While this diversity successfully bridges conceptual gaps (e.g., `PartialEquiv` in Example 2), it introduces distractions if the model lacks the necessary scaffolding to navigate these broader definitions.

**Example 1: `Equiv.Perm.toPartialPerm_inv` (Claude-Opus-4)**

The Problem: Prove that the inverse of a permutation, when converted to a partial permutation, is equal to the inverse of the partial permutation obtained from the original.

Gold Statement:

```
@[simp]theorem thm_P {α : Type u_1} (π : Equiv.Perm α)
    :Equiv.Perm.toPartialPerm π⁻¹ = PartialPerm.symm
    (Equiv.Perm.toPartialPerm π) := by sorry
```

Base Retriever (Success): The Base Retriever successfully found `PartialPerm.toPartialEquiv` (ranked 2nd), which helped the model infer the correct structure.

```
theorem Equiv.Perm.toPartialPerm_inv {α : Type*} (f : Equiv.Perm α) :
    f⁻¹.toPartialPerm = f.toPartialPerm.symm := by sorry
```

Key retrieved premises:

```
Equiv.Perm.toPartialPerm
PartialPerm.toPartialEquiv
PartialPerm.refl
PartialPerm.symm
PartialPerm
```

DRIFT Premises Only (failed typecheck): The model retrieved `PartialPerm.toPartialEquiv` (ranked 3rd) but failed to understand how to use it to bridge `Equiv.Perm` and `PartialPerm.symm`. Instead, it hallucinated an invalid usage $\pi$.`symm.toPartialPerm`.

```
theorem Equiv.Perm.toPartialPerm_inv {α : Type*} (π : Equiv.Perm α) : π
    .symm.toPartialPerm = π.toPartialPerm.symm := by sorry
```

Key retrieved premises:

```
Equiv.Perm.toPartialPerm
PartialPerm.symm
PartialPerm.toPartialEquiv
PartialPerm
Equiv.Perm
```

Full DRIFT (Success): Full DRIFT succeeded because the retrieved theorems explicitly demonstrated the pattern of commuting operations (moving `symm` across `toPartialEquiv`), acting as a structural scaffold.

```
theorem Equiv.Perm.toPartialPerm_inv {α : Type u_1} (π : Equiv.Perm α) :
    (π⁻¹).toPartialPerm = (π.toPartialPerm).symm := by sorry
```

Key retrieved theorems:

```
theorem toPartialEquiv_symm : π.symm.toPartialEquiv = π
    .toPartialEquiv.symm := rfl
```

---

### Example 2: `PartialPerm.coe_toPartialEquiv_symm` (gpt-4.1)

The Problem: The theorem states that for a partial permutation $\pi$, the inverse of the partial equivalence obtained from $\pi$ is equal to the function representing the inverse of $\pi$.

Gold Statement:

```
@[simp]theorem thm_P {α : Type u_1} (π : PartialPerm α)
    :↑(PartialEquiv.symm (PartialPerm.toPartialEquiv π)) =
    (PartialPerm.symm π).toFun := by sorry
```

Base Retriever (Success): Base Retriever found helper lemmas like `invFun_as_coe` and `toFun_as_coe`, which directly link the function coercion to the inverse, guiding the model to a correct formulation using `invFun`.

```
theorem PartialPerm.coe_toPartialEquiv_symm {α : Type*} (π : PartialPerm
    α) : ((π.toPartialEquiv).symm : Part (α → α)) = π.invFun := by sorry
```

Key retrieved premises:

```
Equiv.Perm.toPartialPerm
PartialPerm.toPartialEquiv
PartialPerm.invFun_as_coe
PartialPerm.symm
PartialPerm.toFun_as_coe
```

DRIFT Premises Only (failed typecheck): The model retrieved the necessary definitions but failed to apply them correctly. It attempted to apply `PartialEquiv.symm` directly to $\pi$ (a `PartialPerm`) without converting it first, resulting in a type error.

```
theorem PartialPerm.coe_toPartialEquiv_symm {α : Type*} (π : PartialPerm
    α) : (PartialPerm.symm π).toFun = (PartialEquiv.symm π).toFun := by
    sorry
```

Key retrieved premises:

```
Equiv.Perm.toPartialPerm
Equiv.toPartialEquiv
PartialEquiv.symm
PartialPerm.symm
PartialPerm
```

Full DRIFT (Success): Full DRIFT retrieved the theorem `toPartialEquiv_symm`, which provides an exact template for the equality between the symmetric partial equivalence and the partial equivalence of the symmetric permutation. This scaffold allowed the model to construct a correct statement.

```
theorem PartialPerm.coe_toPartialEquiv_symm {α : Type*} (π : PartialPerm
    α) : (PartialPerm.toPartialEquiv π).symm =
    PartialPerm.toPartialEquiv (PartialPerm.symm π) := by sorry
```

Key retrieved theorems:

```
theorem toPartialEquiv_symm : π.symm.toPartialEquiv = π
    .toPartialEquiv.symm := rfl
```

---

### Example 3: `Cardinal.nonempty_compl_of_mk_lt_mk` (Claude-Opus-4)

The Problem: If the cardinality of set $s$ is less than the cardinality of type $\alpha$, then the complement $s^c$ is nonempty.

Gold Statement:

```
theorem thm_P {α : Type u} {s : Set α} (h : Cardinal.mk ↑s < Cardinal.mk
    α) :Set.Nonempty sᶜ := by sorry
```

Base Retriever (Success): Base Retriever correctly identified and retrieved the function `Cardinal.mk`, enabling the model to apply the premise correctly.

```
theorem Cardinal.nonempty_compl_of_mk_lt_mk {α : Type*} {s : Set α} (h :
    Cardinal.mk s < Cardinal.mk α) : (sᶜ).Nonempty := by sorry
```

Key retrieved premises:

```
Set.Nonempty
Cardinal.mk
Cardinal
Cardinal.IsRegular
Cardinal.aleph0
```

DRIFT Premises Only (failed typecheck): Despite retrieving `Cardinal.mk` (ranked 2nd, same as Base), the model failed to prioritize it over the notation `#s`. The use of `#s < #α` without opening the necessary namespaces caused a typecheck failure.

```
theorem Cardinal.nonempty_compl_of_mk_lt_mk {α : Type*} {s : Set α} (h :
    #s < #α) : (sᶜ).Nonempty := by sorry
```

Key retrieved premises:

```
Set.Nonempty
Cardinal.mk
Set
```

Full DRIFT (Success): Full DRIFT succeeded because it retrieved the theorem `ConNF.`$\mu$`_le_mk_cloud`, which explicitly uses `Cardinal.mk`. This scaffolding guided the model to prefer the explicit function over the notation.

```
theorem Cardinal.nonempty_compl_of_mk_lt_mk {α : Type*} {s : Set α} (h :
    Cardinal.mk s < Cardinal.mk α) : Set.Nonempty sᶜ := by sorry
```

Key retrieved theorems:

```
theorem μ_le_mk_cloud : s.Nonempty → #μ ≤ #(cloud hγβ s) := by
  rintro ⟨t, ht⟩
  refine' (Cardinal.mk_le_mk_of_subset <| subset_cloud ht).trans_eq' _
  rw [Cardinal.mk_image_eq, mk_localCardinal]
  exact typedNearLitter.inj'
```

## A.4 PROMPT TEMPLATES

### A.4.1 DECOMPOSITION PROMPT

This appendix contains the complete prompt used to decompose informal mathematical statements into retrieval queries (the **Decompose** module). It is composed of two parts: a system prompt that defines the model's expert persona and overall task, and a user prompt template that structures the specific input and desired output format.

---

**System Prompt**

```
You are an expert in formal mathematics. Your task is to decompose
    an informal mathematical statement into a set of natural
    language queries. These queries are for retrieving the precise
    definitions, theorems, and structures from a formal mathematics
    library (like mathlib) that are necessary to **formalize** the
    statement.

Your response must be in LaTeX. Decompose the statement into a
    list of queries, with each query enclosed in a '\\boxed{{}}'
    command. The goal is to identify the building blocks for
    writing the statement formally, **not** to find a proof.

You need to:
1. Analyze the informal statement and identify its key
    mathematical components that need formal definitions
2. Break down the statement into natural language queries that
    describe the mathematical concepts and structures needed for
    formalization
3. from the best of your knowledge, come up with the Lean
    representation of it.
4. Focus on what needs to be defined and the implicit hypothesis.
```

---

**User Prompt Template (Instruction and Context)**

```
Given the **Informal statement**, decompose the informal statement
    into retrieval queries for **formalizing** (not proving) the
    statement. Each query must:
- Describe mathematical definitions, structures, or concepts
    needed to formally express the statement in Lean 4
- Explain what mathematical objects or type signatures are involved
- Read as a complete sentence that teaches about the formal
    mathematical structure
- Focus on how to represent concepts formally rather than how to
    prove them
```

```
- Sound like an excerpt from a mathematics reference that explains
    formal definitions

**Important**:
- The goal is FORMALIZATION (translating to Lean 4), NOT finding
    proof strategies
- Write informative descriptions that explain formal concepts and
    definitions
- Each query should describe mathematical structures or type
    information
- Avoid interrogative words (what, how, when, why, etc.)
- The queries should collectively cover all definitions and
    structures needed to write the formal statement

Please return each query using the \\boxed{{}} LaTeX command.

{few_shot_examples}

---
**Informal statement**:
{informal_statement}

**Decomposed queries for formalization**:
```

### A.4.2 FORMALIZATION PROMPT

This appendix contains the complete prompt used to formalize informal mathematical statements into formal statements (the **Formalize Theorems** module). It is composed of two parts: a system prompt that defines the model's expert persona and overall task, and a user prompt template that structures the specific input and desired output format.

**System Prompt**

```
You are an advanced assistant specializing in formal mathematics
    and Lean 4 theorem proving. You have extensive expertise in
    translating mathematical concepts from natural language into
    precise Lean 4 code. Please make sure the generated Lean 4 code
    compiles with {libraries} and Lean version {lean_version}.
```

**User Prompt Template (Instruction and Context)**

```
Given the potential dependent premises listed under **Potential
    dependent premises** (some may be irrelevant) and the
    demonstration examples under **Demonstration examples**,
    translate the natural language statement provided under **
    Informal statement** into a formal Lean 4 theorem. Use the
    theorem name specified under **Name** as the Lean identifier.

Your response must:

- Write only valid Lean 4 code with clear and idiomatic use of
    Lean syntax and conventions
- Include only the formalization - do not include any headers,
    explanations, or proofs
- Use the provided name as the theorem identifier, ensuring it
    adheres to Lean's naming conventions (no hyphens, prefer
    snake_case or camelCase)
```

```
  – Faithfully capture the meaning of the informal statement, paying
      close attention to:
      · Predicate usage and logical structure
      · Type class inference
      · Quantifier scope and binding
      · Mathematical notation and operations
- Enclose all code within triple backticks with the `lean`
      language identifier

Expected Output Format:
```lean
theorem [NAME] : [Lean formalization of the statement] := by sorry
```

Guidelines:
- Select only the relevant premises from those provided
- Ensure proper type annotations where necessary
- Use standard Lean 4 mathematical library conventions, Lean
      version {lean_version}
- Maintain logical equivalence with the informal statement
- Keep the formalization as clean and readable as possible

**Potential dependent premises**
{dependent_premises_list}

**Demonstration examples**
{theorems_list}

**Name**
{problem_full_name}

**Informal statement**
{problem_informal_statement}
```

