# OpenReview forum: "DRIFT: Decompose, Retrieve, Illustrate, then Formalize Theorems"
_ICLR.cc/2026/Conference — ICLR 2026 Poster_

### Official Review · Reviewer_AXr5 · 2025-10-29

**Soundness:** 3
**Presentation:** 3
**Contribution:** 2
**Rating:** 4
**Confidence:** 4

**Summary:**

The paper proposes an end-to-end autoformalization pipeline that converts informal mathematical statements into Lean 4 declarations. The system couples a retriever (trained by the authors) with a generator (commercial or open LLMs) and evaluates outputs using automated Lean checks and a symbolic-equivalence metric. The main claims are (i) training a task-specific retriever improves grounding for generation, and (ii) the pipeline can robustly produce syntactically valid, type-checked Lean statements across a curated benchmark. The paper reports head-to-head comparisons against several general-purpose LLMs and ablations around retrieval.

**Strengths:**

- Clear end-to-end systemization of a practical autoformalization pipeline for Lean 4, with sensible stages (retrieve → generate → check). The paper’s “illustrate” section is genuinely helpful: the qualitative visual walkthroughs of retrieved contexts, intermediate rewrites, and final Lean artifacts improve interpretability and reproducibility.
- Ablations on retrieval are valuable; retrieval quality is a major bottleneck in Lean autoformalization, so any careful analysis there is welcome.
- Timely topic & potential impact. There is fast-moving SOTA on Lean autoformalization and proof generation (Goedel/ Kimina/ process-supervised approaches); a robust, reproducible pipeline can help the community measure progress on statements specifically.

**Weaknesses:**

W1. Retriever novelty & baselines are weak.

The only trained component is a DPR-style dual encoder, which is basically the same as one in RAutouformalizer. Moreover, Lean-specific retrieval methods already exist (LeanSearch’s semantic search[3]; LeanExplore’s hybrid multi-signal retrieval[4]), and the paper neither compares against nor leverages them as baselines or components. This makes it hard to justify training a new DPR when stronger plug-ins are available.

W2. Missing SOTA autoformalizer baselines.

 The paper compares primarily to generic LLMs (GPT-4.1, DeepSeek-V3.1) but omits direct comparisons to autoformalizer-specialized systems, notably Goedel-Formalizer[1] and Kimina-Autoformalizer[2], which are expressly designed to translate informal math to Lean 4 statements and are publicly available. Given their focus and reported quality, they are the most relevant baselines for this task. Including them (or explaining why they cannot be included) is essential for positioning.

W3. Frontier model coverage is incomplete/out-of-date.

For the generator, the paper focuses on GPT-4.1 and DeepSeek V3.1. The current frontier for mathematical/logic tasks prominently features DeepSeek R1 (0528), OpenAI’s o-series (o3), Gemini 2.5 Pro, and Claude 4.1; these models publicly advertise stronger reasoning/coding capabilities and should be part of the comparison, at least in a retrieval-on vs retrieval-off ablation to substantiate the retriever’s benefit. Having only a single frontier model (Claude 4) is not persuasive in 2025.

W4. Stale toolchain / dataset snapshot raises representativeness concerns.

 Experiments are run on Lean 4.7.0 / an older mathlib snapshot. Lean and mathlib have evolved significantly (Lean 4.25.0-rc2 exists; mathlib’s scale has expanded beyond 200k theorems), and style/namespace changes accumulate. Results confined to an older snapshot may under- or over-estimate real-world robustness. Authors should either (i) re-run on a contemporary toolchain and a recent mathlib commit or (ii) justify the choice and discuss compatibility gaps.

W5. Evaluation metric (BEq+) may under-state performance without human adjudication.

 BEq+ is a reasonable automated proxy, but even its authors note a relatively high false-negative rate; strict symbol-level equivalence can mark semantically correct paraphrases as wrong. The paper reports low success rates (sub-25% in places); without a human-adjudicated subset or complementary metrics, it is hard to interpret practical significance. A small-scale human study or relaxed-equivalence cross-check (e.g., type-equivalence under definitional unfolding) would strengthen claims.

[1] Yong Lin, et al. "Goedel-Prover-V2: Scaling Formal Theorem Proving with Scaffolded Data Synthesis and Self-Correction" arXiv preprint arXiv:2508.03613 (2025)

[2] Wang, Haiming, et al. "Kimina-Prover Preview: Towards Large Formal Reasoning Models with Reinforcement Learning" arXiv preprint arXiv:2504.11354 (2025).

[3] Gao, Guoxiong, et al. "A semantic search engine for Mathlib4." arXiv preprint arXiv:2403.13310 (2024).

[4] Asher, Justin. "LeanExplore: A search engine for Lean 4 declarations." arXiv preprint arXiv:2506.11085 (2025).

**Questions:**

Toolchain & dataset. What constraints led you to Lean v4.7.0/that mathlib commit? Please discuss how brittle your pipeline is to syntax/tactic drift across versions. Additionally, could you explain in detail how you conduct data extraction from mathlib and prepare them for embedding training in detail?

Ground truth. How do you construct ground truth (oracles) for decomposition and retrieval tasks? Detail the pipeline used to obtain ground truth (oracles) from Lean.

Interpreting BEq+. Given BEq+’s known false negatives, do you have a human-adjudicated subset to calibrate precision/recall? How often do your “failures” reflect symbol-level mismatches vs true semantic errors? Consider reporting: (a) case study on type-checks but BEq+-fails; (b) human-judged correctness.

Ablations on retrieval → generation sensitivity. Please report end-to-end success vs top-k retrieval quality (e.g., R@k buckets) to quantify how much the generator depends on retrieval depth and filtering.

---

> ### Author Response · Authors · 2025-11-21
> **Response to Reviewer AXr5 (1/3)**
>
> > Retriever novelty & baselines are weak. The only trained component is a DPR-style dual encoder, which is basically the same as one in RAutouformalizer. Moreover, Lean-specific retrieval methods already exist (LeanSearch’s semantic search[3]; LeanExplore’s hybrid multi-signal retrieval[4]), and the paper neither compares against nor leverages them as baselines or components. This makes it hard to justify training a new DPR when stronger plug-ins are available.
>
> We fear there may be a misunderstanding concerning our contribution and task. We respectfully clarify that **our primary contribution is not the DPR-style dual encoder but rather the DRIFT pipeline itself: decomposing informal statements into granular sub-queries, leveraging the LLM's Lean knowledge to guide retrieval for each component, and providing illustrative theorems alongside retrieved premises. These design choices are novel and directly impact performance**, as demonstrated in Table 3 where each component provides performance improvements.
>
> We kindly refer to the comments from other reviewers that recognized these architectural innovations as the core contribution:
>
> - **Reviewer Wmuf** highlighted that the Illustrate step addresses the "gap between definition and usage, an underexplored angle in prior work," describing the end-to-end design as "clear" and "validated."
> - **Reviewer nbLa** confirmed that "the pipeline for autoformalization is novel and has not been explored before," noting that the modular techniques are "interesting" and provide valuable insights via ablation.
> - **Reviewer 3Kvp** acknowledged that through this "structured process," DRIFT effectively decomposes complex statements to guide correct application.
>
> We hope this clarifies that our novelty lies in the structural redesign of the retrieval process to align with autoformalization, rather than in the standard training of the dense retriever component.
>
> Regarding LeanSearch and LeanExplore, these systems are designed for concept-based theorem retrieval (e.g., "find theorems about continuity"), whereas DRIFT addresses a fundamentally different task: decomposing complex informal mathematical statements into multiple components and retrieving targeted premises for each. **The retrieval model serves as a component within our pipeline, but the core innovation lies in how we structure the retrieval process to align with the autoformalization task.**
>
> **Actions taken:** We extended the related work discussion (Section 2) to distinguish DRIFT from other retrieval methods.
>
> > Missing SOTA autoformalizer baselines. The paper compares primarily to generic LLMs (GPT-4.1, DeepSeek-V3.1) but omits direct comparisons to autoformalizer-specialized systems, notably Goedel-Formalizer[1] and Kimina-Autoformalizer[2], which are expressly designed to translate informal math to Lean 4 statements and are publicly available. Given their focus and reported quality, they are the most relevant baselines for this task. Including them (or explaining why they cannot be included) is essential for positioning.
>
> We thank the reviewer for suggesting these specific baselines. **We excluded Kimina-Autoformalizer because its training data explicitly includes ProofNet and MiniF2F (Kimina-Prover[1], Appendix C.2), rendering evaluation on these benchmarks unfair due to data contamination.**
> While Goedel is effective in standard in-distribution translation, we hypothesize that it may lack the robustness for out-of-distribution tasks like ConNF. *As requested, we are currently benchmarking the Goedel-Formalizer-V2 (7B) model on our datasets and will update the results before the end of rebuttal.*
>
> **Actions taken:** We evaluate Goedel-Formalizer and compare the performance with DRIFT. We will update the results before the end of the rebuttal.
>
> References:
>
> - [1] Wang, Haiming, et al. "Kimina-prover preview: Towards large formal reasoning models with reinforcement learning." arXiv preprint arXiv:2504.11354 (2025).

---

> > ### Author Response · Authors · 2025-11-21
> > **Response to Reviewer AXr5 (2/3)**
> >
> > > Frontier model coverage is incomplete/out-of-date. For the generator, the paper focuses on GPT-4.1 and DeepSeek V3.1. The current frontier for mathematical/logic tasks prominently features DeepSeek R1 (0528), OpenAI’s o-series (o3), Gemini 2.5 Pro, and Claude 4.1; these models publicly advertise stronger reasoning/coding capabilities and should be part of the comparison, at least in a retrieval-on vs retrieval-off ablation to substantiate the retriever’s benefit. Having only a single frontier model (Claude 4) is not persuasive in 2025.
> >
> > **We respectively disagree with the characterization that our model coverage is incomplete/out-of-date. Our model selection (GPT-4.1, DeepSeek-V3.1, and Claude-Opus-4) provides a representative assessment of frontier capabilities, a choice validated by both empirical benchmarks and recent literature**. For instance, DeepSeek-Prover-V2 [2] utilized DeepSeek-V3 for sub-goal decomposition and proof step formalization, Goedel-Prover-V2 [3] relied on Claude-Sonnet-4 for initializing expert iteration, Kimini-Prover [4] synthesized dataset using Claude-Sonnet-3.7. This confirms that our selected models represent the standard, high-capability reasoning engines used for complex formalization tasks in the current research landscape.
> >
> > While the field moves rapidly, evidenced by the release of Gemini 3 and GPT-5.1 subsequent to the reviewer’s comments, our results establish a model-agnostic trend of Decomposition Robustness. We refer the reviewer to Table 1 (Section 5.1) and Table 8 (Section A.2.7), models perform comparably as Decomposers despite significant variations in their formalization strength observed in Table 2 (Section 5.2). This highlights the fundamental value of the DRIFT framework where it enables models to leverage their strong natural language understanding to secure precise retrieval and bridge the formalization gap.
> >
> > Notably, DeepSeek-V3.1 (used in our study) was released after the DeepSeek-R1-0528 snapshot and empirically surpasses it on coding benchmarks.
> >
> > References:
> >
> > - [2] Ren, Z. Z., et al. "Deepseek-prover-v2: Advancing formal mathematical reasoning via reinforcement learning for subgoal decomposition." arXiv preprint arXiv:2504.21801 (2025).
> > - [3] Lin, Yong, et al. "Goedel-prover-v2: Scaling formal theorem proving with scaffolded data synthesis and self-correction." arXiv preprint arXiv:2508.03613 (2025).
> > - [4] Wang, Haiming, et al. "Kimina-prover preview: Towards large formal reasoning models with reinforcement learning." arXiv preprint arXiv:2504.11354 (2025).
> >
> > > Stale toolchain / dataset snapshot raises representativeness concerns. Experiments are run on Lean 4.7.0 / an older mathlib snapshot. Lean and mathlib have evolved significantly (Lean 4.25.0-rc2 exists; mathlib’s scale has expanded beyond 200k theorems), and style/namespace changes accumulate. Results confined to an older snapshot may under- or over-estimate real-world robustness. Authors should either (i) re-run on a contemporary toolchain and a recent mathlib commit or (ii) justify the choice and discuss compatibility gaps.
> >
> > **We utilize distinct Lean versions necessitated by data constraints and benchmark requirements.** For retriever training, we utilized the training data of the retriever from RAutoformalizer, which provides the necessary aligned informal-formal pairs on Mathlib 4.7.0. During inference, valuation of the generated formal statement, we evaluated using Lean 4.18.0 (released April 2025) for ProofNet and MiniF2F. The high compilation success rate of the gold statements (~100%) confirms that the version difference between the retriever's training data and the evaluation environment does not introduce instability. For evaluation of ConNF formalization, we utilized Lean 4.7.0 strictly because the benchmark is pinned to this version by its dependencies. Crucially, these environments are consistent across all baselines and datasets in our study. There is no systematic bias that would artificially favor DRIFT over Zero-Shot or RAuto; any version-related friction applies equally to all methods.
> >
> > **Our version selection aligns with established practices in the autoformalization community. Recent studies including Goedel-Prover and Kimina-Prover similarly pinned their evaluations to specific versions (e.g., Lean 4.9.0) to align with previous work (DeepSeek-Prover-V1.5).**
> >
> > Regarding brittleness, we emphasize that DRIFT is inherently more robust and efficient than fine-tuning. Because DRIFT retrieves definitions from the active library context, adapting to a new Lean version or Mathlib release requires only re-indexing the library (a low-cost offline step). In contrast, fine-tuned models memorize static syntax and require expensive retraining to adapt to deprecated or renamed theorems.
> >
> > **Actions taken:** We added the Toolchain Versions and Robustness discussion in Appendix A.1.2.

---

> > > ### Author Response · Authors · 2025-11-21
> > > **Response to Reviewer AXr5 (3/3)**
> > >
> > > > Evaluation metric (BEq+) may under-state performance without human adjudication. BEq+ is a reasonable automated proxy, but even its authors note a relatively high false-negative rate; [...]. A small-scale human study or relaxed-equivalence cross-check (e.g., type-equivalence under definitional unfolding) would strengthen claims. Q3: Interpreting BEq+. Given BEq+’s known false negatives, do you have a human-adjudicated subset to calibrate precision/recall? How often do your “failures” reflect symbol-level mismatches vs true semantic errors?[...]
> > >
> > > We acknowledge that BEq+ is a strict metric with a known false-negative rate. **However, this strictness applies systematically across all baselines, ensuring that the relative improvements we report (e.g., >50% gain on ConNF) are robust and not artifacts of the metric**. BEq+ likely underestimates absolute success but accurately reflects the comparative efficacy of DRIFT. We prioritized this rigorous symbolic verification over "LLM-as-a-judge" to avoid the well-documented risks of hallucination and self-preference bias in LLM adjudication. While refining equivalence metrics is a valuable community goal, it remains outside the scope of this framework contribution.
> > >
> > > > Toolchain & dataset. What constraints led you to Lean v4.7.0/that mathlib commit? Please discuss how brittle your pipeline is to syntax/tactic drift across versions. Additionally, could you explain in detail how you conduct data extraction from mathlib and prepare them for embedding training in detail?
> > >
> > > Regarding data extraction and training, we emphasize that training the retriever is not our primary contribution; first, **we adhered to the protocol and utilized the informalized Mathlib data from RAutoformalizer [5,6]**. Specifically, we fine-tuned the BGE-M3 retriever using the FlagEmbedding library (hyperparameters in Table 4, Section A.1.2). We employed a standard contrastive loss where the informal statement serves as the anchor, the formal declaration of the ground truth dependent premises serve as positive labels, and other random library samples serve as negative labels. Crucially, the retriever was trained exclusively on Mathlib data to ensure strict isolation from the test sets and to preserve ConNF as a valid OOD benchmark.
> > >
> > > **Actions taken:** We provided additional details in Appendix A.1.2.
> > >
> > > > Ground truth. How do you construct ground truth (oracles) for decomposition and retrieval tasks? Detail the pipeline used to obtain ground truth (oracles) from Lean.
> > >
> > > **There is no defined “ground truth” for decomposition, the decompose module is evaluated through the downstream “Dependency Retrieval”**. The ground-truth dependencies for ProofNet and ConNF are the benchmark data released by RAutoformalizer. For MiniF2F, we constructed ground truth dependencies following RAutoformalizer using Jixia [7], a static analysis tool for Lean 4 that parses the Abstract Syntax Tree to identify the precise set of source-level declarations required for compilation.
> > >
> > > **Actions taken:** We added additional clarification in Appendix A.1.2 and added the data extraction details in Appendix A.1.2.
> > >
> > > > Ablations on retrieval → generation sensitivity. Please report end-to-end success vs top-k retrieval quality (e.g., R@k buckets) to quantify how much the generator depends on retrieval depth and filtering.
> > >
> > > Regarding "success vs. top-k" analysis, DRIFT’s design renders standard monolithic buckets inapplicable. **Instead of pulling a large ranked list, DRIFT generates sub-queries adaptively (avg. 5.77) and retrieves the top-1 result for each to enforce atomic mapping between informal concepts and formal definitions**. Increasing k per sub-query is detrimental because the target atomic dependency is typically unique; retrieving multiple candidates disproportionately amplifies noise. **Thus, retrieval "depth" is defined not by rank, but by the granularity of decomposition, which scales autonomously with problem complexity (e.g., 6.42 sub-queries for ConNF vs. 5.21 for MiniF2F, details in Table 8, A.2.7).**
> > >
> > > We respectfully clarify that training a task-specific retriever is not our primary contribution; we follow the RAuto protocol for that component. Our core novelty lies in the DRIFT framework: specifically, the decomposition-driven retrieval strategy (transforming monolithic queries into atomic concepts) and the Illustration mechanism (providing syntactic scaffolding).
> > >
> > > **Actions taken:** We revised our discussion of the contributions (Section 1) and highlighted that the number of sub-queries and premises retrieved by DRIFT is autonomously determined by the Decomposer.
> > >
> > > References:
> > >
> > > - [5] Liu, Qi, et al. "Rethinking and improving autoformalization: towards a faithful metric and a dependency retrieval-based approach." The Thirteenth International Conference on Learning Representations. 2025.
> > > - [6] Data at: https://github.com/Purewhite2019/rethinking_autoformalization
> > > - [7] Jixia Library: https://github.com/frenzymath/jixia

---

### Official Review · Reviewer_3Kvp · 2025-10-29

**Soundness:** 3
**Presentation:** 3
**Contribution:** 3
**Rating:** 6
**Confidence:** 3

**Summary:**

This paper proposes the DRIFT framework , which aims to enhance the process of mathematical statement autoformalization. DRIFT decomposes complex informal mathematical statements into sub-queries, then accurately retrieves the corresponding formal dependencies. Based on these retrieved dependencies, it provides illustrative contextual examples to guide the model in applying them correctly. Through this structured process, DRIFT achieves improved performance in automatic formalization of mathematical statements.

**Strengths:**

1.DRIFT introduces an innovative approach by decomposing complex informal statements into sub-queries, which allows for precise retrieval of the required formal dependencies. In addition to retrieving relevant premises, the framework also provides illustrative examples of their usage, effectively guiding the model to apply the dependencies correctly during formalization.

2.The experiments demonstrate both the generality and effectiveness of the proposed framework. Moreover, the ablation studies clearly reveal the individual contributions and roles of different modules within DRIFT, strengthening the empirical support for the proposed design.

**Weaknesses:**

1.The ablation study shows that the retrieval module plays a crucial role in the overall performance of DRIFT. However, the paper does not compare this module with existing Lean premise retrieval methods, such as Lean Search or other established retrieval baselines. Including such comparisons would provide a clearer understanding of the advantages and limitations of the proposed retrieval component.

2.The experiments primarily focus on general-purpose reasoning models such as GPT and DeepSeek. However, there are now several large models that have been specifically trained or fine-tuned on Lean. It remains unclear whether DRIFT has been tested on these Lean-specialized models, and such evaluation could further demonstrate the framework’s adaptability and robustness.

**Questions:**

Please refer to the Weakness section.

---

> ### Author Response · Authors · 2025-11-21
> **Response to Reviewer 3Kvp**
>
> > The ablation study shows that the retrieval module plays a crucial role in the overall performance of DRIFT. However, the paper does not compare this module with existing Lean premise retrieval methods, such as Lean Search or other established retrieval baselines. Including such comparisons would provide a clearer understanding of the advantages and limitations of the proposed retrieval component.
>
> **We distinguish our task from Theorem Retrieval tools like LeanSearch, Moogle, and BM25, which identify semantically similar theorems for proof assistance. In contrast, DRIFT and RAuto target dependency retrieval**: pinpointing the precise formal definitions (e.g., IsPrimitiveRoot, ZMod) required for code compilation. Consequently, we strictly benchmark against RAuto (DPR), the established SOTA for this specific task.
> As shown in Table R2, our approach differs fundamentally in both query and retrieval strategy. While baselines use query augmentation or keywords to find analogous theorems, DRIFT employs Atomic Decomposition to map specific concepts. Furthermore, our "Theorem Selection" (Illustrate) is conditional on retrieved dependencies, explicitly demonstrating syntactic usage rather than selecting theorems based on semantic similarity.
>
> | Feature              | DRIFT (Ours)                                 | RAuto (Liu et al., 2025)              | LeanSearch / Moogle                  | BM25 (Standard)                |
> |----------------------|----------------------------------------------|---------------------------------------|--------------------------------------|--------------------------------|
> | Target Task          | Dependency Retrieval                         | Dependency Retrieval                  | Theorem Retrieval                    | Theorem Retrieval              |
> | Goal                 | Find precise definitions/types               | Find precise definitions/types        | Find similar/relevant theorems       | Find theorems                  |
> | Query Strategy       | Atomic Decomposition (Sub-queries)           | Monolithic Informal Statement         | Query Augmentation (Explanation)     | Keywords / Bag-of-Words        |
> | Dependency Retrieval |Sub-query to Definition     | Monolithic (Statement to Definitions) | N/A (Implicit)                       | N/A                            |
> | Theorem Selection    | Usage-Based (Demonstrate retrieved premises) | N/A                                   | Similarity-Based (Semantic Distance) | Keyword-Based (Term Frequency) |
>
> **Actions taken:** We extended the related work section (Section 2) in the revised manuscript.
>
> > The experiments primarily focus on general-purpose reasoning models such as GPT and DeepSeek. However, there are now several large models that have been specifically trained or fine-tuned on Lean. It remains unclear whether DRIFT has been tested on these Lean-specialized models, and such evaluation could further demonstrate the framework’s adaptability and robustness.
>
> We prioritized frontier models (DeepSeek-V3.1, GPT-4.1) over smaller fine-tuned specialists because **DRIFT is a reasoning pipeline requiring advanced instruction-following robustness**. Crucially, our preliminary experiments revealed that finetuned models suffer from overfitting and a lack of generalizability. While these models were trained with retrieval augmentation, they were specialized to the specific content of the retriever used during their training. When presented with DRIFT's distinct context structure (e.g., atomic premises and illustrative theorems), these models failed to generalize to the distributional shift, struggling to utilize the new format effectively. In contrast, frontier models possess the general reasoning robustness required to adapt to diverse, out-of-distribution retrieval contexts, a capability critical for the OOD generalization (ConNF) results demonstrated in our study.

---

> > ### Comment · Reviewer_3Kvp · 2025-11-26
> > **Official Comment by Reviewer 3Kvp**
> >
> > Thanks for the rebuttal. The authors have effectively addressed my concerns. I remain positive about this work.

---

### Official Review · Reviewer_Wmuf · 2025-11-01

**Soundness:** 3
**Presentation:** 3
**Contribution:** 3
**Rating:** 6
**Confidence:** 4

**Summary:**

The paper proposes DRIFT: a four-stage framework for autoformalizing informal math statements into Lean. DRIFT (i) Decomposes an informal statement into atomic, concept-focused sub-queries (with predicted formal “anchors”), (ii) Retrieves dependent premises from a formal library using a finetuned dense retriever, (iii) Illustrates usage via a small set of demonstrative theorems chosen by a greedy coverage algorithm, and (iv) Formalizes the statement conditioned on the retrieved context. Across ProofNet (in-distribution), MiniF2F-test (largely self-contained), and ConNF (out-of-distribution), DRIFT improves dependency retrieval F1 and downstream formalization, with especially large gains on ConNF where it even surpasses an oracle* retrieval setting.

**Strengths:**

1. The framework adds an Illustrate step that selects a minimal set of theorems to demonstrate how retrieved premises are used, addressing the gap between definition and usage, an underexplored angle in prior work.

2. Clear end-to-end design validated on three complementary benchmarks; the method substantially boosts BEq+ and type-check rates over strong retrieval baselines and zero-shot, with striking OOD gains on ConNF.

3. Ablations show the Illustrate step is crucial (removing it sharply reduces BEq+ on ProofNet/ConNF), and quantify contributions of Decompose vs. Retrieval.

4. The pipeline is easy to follow, and can inform future RAG design for formal methods.

**Weaknesses:**

1. The decomposer appends predicted formal representations; the paper argues this helps anchoring, but does not quantify robustness when anchors are wrong/noisy.

2. The ablation discussion suggests decomposed retrieval may introduce diverse noise that requires illustrative scaffolding; more error taxonomy and qualitative failure analyses (on both ProofNet and MiniF2F) would bolster the explanation and guide adaptive retrieval strategies.

3. There is a chance that I missed this somewhere, but the paper does not seem to report compute/latency costs for decomposition + retrieval + illustration vs. baselines? Given practical adoption, cost-quality tradeoffs matter.

4. The paper observes retrieval can distract in low-dependency regimes (MiniF2F). It would help to show an adaptive gate (e.g., predicting when to skip retrieval or to down-weight illustration) and quantify decision quality.

**Questions:**

1. How does performance vary with the number of sub-queries, top-k per sub-query, and the illustration budget m? Any evidence of diminishing returns or overfitting with larger m?

2. You surpass Oracle* on ConNF; can you provide diagnostics (e.g., premise coverage of selected theorems, overlap with ground truth, qualitative examples) explaining where illustrative theorems help beyond oracle dependencies?

3. Some slight inconsistencies in terminologies: e.g., "DeepSeek-3.1" and "DeepSeek-V3.1" both appeared.

---

> ### Author Response · Authors · 2025-11-21
> **Response to Reviewer Wmuf (1/2)**
>
> > The decomposer appends predicted formal representations; the paper argues this helps anchoring, but does not quantify robustness when anchors are wrong/noisy.
>
> Thank you for raising this point. **We kindly refer to Table 7 in Appendix A.2.6 where we evaluate the value of the predicted formal representations for formalization in our Parametric Retrieval ablation and we found that using these predictions directly for formalization yields limited success (e.g., 13.64% BEq+ on ProofNet with GPT-4.1)**. However, our system demonstrates strong robustness to this noise: the Retrieve module consistently achieves high performance (e.g., 26.64% F1 on ProofNet with GPT-4.1 as decomposer). In an additional experiment restricting the decomposer to natural language only, the retrieval F1 performance on ProofNet dropped significantly to 8.88%.
>
> > The ablation discussion suggests decomposed retrieval may introduce diverse noise that requires illustrative scaffolding; more error taxonomy and qualitative failure analyses (on both ProofNet and MiniF2F) would bolster the explanation and guide adaptive retrieval strategies.
>
> To provide a more granular error taxonomy, **we have added a new qualitative error analysis in the Appendix**. This analysis investigates failure cases on MiniF2F where the zero-shot baseline succeeded but DRIFT failed. We found **these failures typically occur in low-complexity problems where retrieval introduced noise**, and we categorized the primary failure modes into variable naming issues (such as undefined variables), over-complication (resulting in typecheck errors), and formalization style mismatches (this category includes examples for false negative evaluations of BEq+).
> Furthermore, to explicitly evaluate the potential for adaptive strategies, we conducted an "Oracle Ensemble" study to measure the theoretical upper bound of dynamically selecting between Zero-Shot and DRIFT. **This adaptive retrieval method consistently outperforms the individual methods (e.g., raising Claude's BEq+@1 from 32.59% to 35.27%)**.
> We demonstrated the effectiveness and importance of illustrative theorems in the qualitative analysis of DRIFT compared to Oracle* on ConNF. We will provide further discussions to the qualitative analysis in the camera-ready version of the manuscript.
>
> **Actions taken:** We provided a detailed qualitative error analysis in the Appendix A.3.1.
>
> > There is a chance that I missed this somewhere, but the paper does not seem to report compute/latency costs for decomposition + retrieval + illustration vs. baselines? Given practical adoption, cost-quality tradeoffs matter.
>
> While DRIFT incurs higher per-inference token costs due to the additional decomposition step and expanded input context (illustrative theorems), along with approximately 6 times more embedding operations, however this yields superior total cost-efficiency compared to zero-shot formalization. On ProofNet (GPT-4.1), a single DRIFT inference (Pass@1: 17.38%) outperforms ten Zero-Shot attempts (Pass@10: 13.37%). **Since DRIFT effectively replaces more than 10 baseline attempts while incurring roughly only 2 times the token and compute cost per inference, it provides a 5x gain in cost-to-solution in practice**.
>
> **Actions taken:** We will add a cost and latency analysis in the camera-ready version of the paper.
>
> > The paper observes retrieval can distract in low-dependency regimes (MiniF2F). It would help to show an adaptive gate (e.g., predicting when to skip retrieval or to down-weight illustration) and quantify decision quality.
>
> **We address the potential of an adaptive gate by calculating the "Oracle Ensemble" performance, the theoretical upper bound of dynamically selecting between Zero-Shot and DRIFT.** As shown below, the ensemble consistently outperforms individual methods (e.g., raising Claude's BEq+@1 from 32.59% to 35.27%), confirming that the approaches are complementary: DRIFT handles complexity while Zero-Shot inference avoids distraction. While training a classifier or adaptive LLMs is outside our scope, these results inspire future work on adaptive retrieval.
>
> Table: "Oracle Ensemble" Performance on MiniF2F (Pass@1).
>
> | Model             | Metric | Zero-Shot | DRIFT  | **Oracle Ensemble** (Adaptive Upper Bound) |
> |-------------------|--------|-----------|--------|--------------------------------------------|
> | **Claude-Opus-4** | TC@1   | 95.09%    | 93.75% | **95.98%**                                 |
> |                   | BEq+@1 | 31.25%    | 32.59% | **35.27%**                                 |
> | **GPT-4.1**       | TC@1   | 69.64%    | 74.55% | **81.70%**                                 |
> |                   | BEq+@1 | 23.21%    | 24.55% | **25.89%**                                 |
>
> **Actions taken:** We added the Potential for Adaptive Retrieval discussion in Appendix A.2.9.

---

> > ### Author Response · Authors · 2025-11-21
> > **Response to Reviewer Wmuf (2/2)**
> >
> > > How does performance vary with the number of sub-queries, top-k per sub-query, and the illustration budget m? Any evidence of diminishing returns or overfitting with larger m?
> >
> > Thank you for the question, we analyze parameter sensitivity based on DRIFT's design principles. For the number of sub-queries, and the top-k per sub-query, we refer you to Section 3.2, Section 4.3, and Appendix A.2.7. First, **the number of sub-queries is not a fixed hyperparameter but is adaptively selected by the Decomposer with problem complexity** (e.g., averaging 5.21 for MiniF2F vs. 6.42 for ConNF, details in Table 8, Section A.2.7), ensuring decomposition granularity aligns with information density. Second, **we set top-k=1 per sub-query to enforce an atomic, one-to-one mapping between queries and formal concepts**; increasing k disproportionately amplifies the volume of irrelevant premises (distractors) relative to the signal, exacerbating the noise issues observed in our ablation on MiniF2F.
> >
> > In response to the theorem illustration budget, we calculated the premise coverage rate change as compared to the theorem top-m (see Figure 2 in Appendix A.2.8).  **As shown in Figure 2, m=3 represents the critical point of diminishing returns (the "elbow" of the curve)**. While coverage marginally increases up to m=5, the slope flattens significantly after m=3, indicating that the vast majority of discoverable premises are captured within the first three theorems. We deliberately stop at this efficient frontier because striving for the final ~5-10% of coverage requires adding more theorems, which disproportionately introduces irrelevant "distractor" premises and token overhead. This decision is supported by our ablation study (Table 3, Section 5.3), where we observed that excessive context degrades performance in low-dependency regimes; thus, capping m at the elbow ensures we capture the necessary signal while minimizing the noise that harms generation.
> >
> > **Actions taken:** We added the empirical analysis of premise coverage rate versus theorem selection budget in Appendix A.2.8.
> >
> > > You surpass Oracle on ConNF; can you provide diagnostics (e.g., premise coverage of selected theorems, overlap with ground truth, qualitative examples) explaining where illustrative theorems help beyond oracle dependencies?
> >
> > We address the phenomenon of DRIFT surpassing the Oracle* baseline through a **detailed qualitative analysis added to Appendix A.3.2.
> > Our qualitative analysis demonstrates that while Oracle provides the correct premises, it fails to provide the recipe for their usage**. In Example 1 (ConNF.Pretangle.ofCoe_inj), the Oracle model failed because it omitted the crucial ConNF namespace, a syntactic convention not inferable from the premise list alone. In contrast, DRIFT retrieved the structurally identical theorem toCoe_inj, which served as a syntactic template, demonstrating the correct namespace usage. Similarly, in Example 3, the Oracle model missed a non-obvious global typeclass instance (ConNF.FOAAssumptions). DRIFT's retrieved theorems provided context that included this dependency. Statistically, we found that around 78% of Oracle failures on ConNF were typecheck errors (often due to missing namespaces or incorrect type coercions) rather than fundamental logic errors. DRIFT mitigates this by providing ‘illustrative theorems’ that act as syntactic scaffolding, revealing implicit arguments, type coercions, and typeclass requirements that a simple list of names cannot convey.
> >
> > **Actions taken:** We provided a detailed qualitative error analysis in Appendix A.3.2.
> >
> > > Some slight inconsistencies in terminologies: e.g., "DeepSeek-3.1" and "DeepSeek-V3.1" both appeared.
> >
> > **Actions taken:** We thank the reviewer for this remark. We have corrected this inconsistency and standardized the terminology to DeepSeek-V3.1 throughout the revised manuscript.

---

> > > ### Comment · Reviewer_Wmuf · 2025-11-25
> > >
> > > Thanks for the detailed responses! I would like to maintain my positive evaluation of this work.

---

### Official Review · Reviewer_nbLa · 2025-11-07

**Soundness:** 3
**Presentation:** 3
**Contribution:** 3
**Rating:** 6
**Confidence:** 4

**Summary:**

The paper proposes DRIFT, a new technique for autoformalization. Their pipeline first decomposes a statement into sub-problems, retrieves relevant premises and their usage in sample theorems, and finally uses all the included retrievals to perform formalization.

**Strengths:**

- The pipeline for autoformalization is novel and has not been explored before in the literature. The results are also strong, and the authors also ablate removing several modules (Sec 5.3), providing interesting insights into the value add of each module.
- The techniques used for the modules are interesting, and the method outperforms zero-shot and retrieval-augmented baselines.

**Weaknesses:**

- As discussed in the paper, I suspect that the benchmarks could be contaminated, especially miniF2F and ProofNet. There are many instances of these two popular benchmarks on GitHub, and it would be surprising if models had not seen them before, even if the results are low. It is possible that retrieval could remind or steer the model to a certain distribution that can elicit its recall ability of seeing these benchmarks.
- The pipeline consists of a set of modules, but none of them seem to be particularly optimized for performance. For example, for the decompose module, only one decomposition prompt seems to have been tested.
- The previous issue could lead to misleading interpretations of the ablation study: the authors noted that in one of the experiments, removing the "decompose" module does not degrade performance, but I wonder if this would be different if the retrieval and formalizer models were replaced with stronger models.
- The paper does not compare with other autoformalization techniques in the literature, making it hard to assess its significance and effectiveness

**Questions:**

- Can the authors demonstrate gains on proof autoformalization using this method as well?
- Why was m=3 selected for the illustration stage? Is it possible that scaling this up to much more examples will improve more (e.g. in https://arxiv.org/abs/2404.11018)
- How does DRIFT compare to other autoformalization techniques in performance?

---

> ### Author Response · Authors · 2025-11-21
> **Response to Reviewer nbLa (1/2)**
>
> > As discussed in the paper, I suspect that the benchmarks could be contaminated, especially miniF2F and ProofNet. There are many instances of these two popular benchmarks on GitHub, and it would be surprising if models had not seen them before, even if the results are low. It is possible that retrieval could remind or steer the model to a certain distribution that can elicit its recall ability of seeing these benchmarks.
>
> We appreciate the reviewer's concern regarding potential data contamination in MiniF2F and ProofNet. **Our Parametric Retrieval ablation study (Table 7, Section A.2.6) provides evidence against the hypothesis that DRIFT merely elicits memorized solutions**. In this ablation, we evaluate formalization using only the model’s parametric knowledge without external retrieval. On ProofNet with GPT-4.1, the parametric-only baseline achieves 13.64% BEq+@1 while full DRIFT achieves 17.38%. If DRIFT were simply triggering recall, we would not observe this gap, as both conditions would have equal access to the model's memory. Additionally, if contamination were the primary factor, we would expect better zero-shot performance.
>
> **Actions taken:** We extend the discussion regarding data contamination in the Appendix A.2.6.
>
> > The pipeline consists of a set of modules, but none of them seem to be particularly optimized for performance. For example, for the decompose module, only one decomposition prompt seems to have been tested.
>
> We emphasize that our primary objective was to **establish a generalizable framework robust across different models, rather than to optimize the approach for specific setups**. Our current decomposition prompt yields consistent sub-query patterns across three distinct frontier models (Table 8, Section A.2.7) and consistently improves retrieval F1 scores across all benchmarks (Table 1, Section 5.1). **We agree with the reviewer that prompt engineering can possibly yield further performance gains for specific LLMs**, but this is beyond the research scope of the paper.
>
> **Actions taken:** We revised the methodology (Section 3) and implementation details (Section 4.3) to highlight our focus on establishing a generalizable DRIFT framework.
>
> > The previous issue could lead to misleading interpretations of the ablation study: the authors noted that in one of the experiments, removing the "decompose" module does not degrade performance, but I wonder if this would be different if the retrieval and formalizer models were replaced with stronger models.
>
> We note that our study employs the strong frontier models. The results reflect the capabilities of the SOTA at the time of writing. The observation that removing “Decompose” has minimal impact on MiniF2F aligns with the benchmark's specific characteristics. **Since MiniF2F is largely self-contained (avg 0.43 dependencies), the benefits of retrieval are therefore limited**.
>
> **Actions taken:** We extend the discussion on retrieval for MiniF2F and provide a qualitative failure analysis in Appendix A.3.1.

---

> > ### Author Response · Authors · 2025-11-21
> > **Response to Reviewer nbLa (2/2)**
> >
> > > The paper does not compare with other autoformalization techniques in the literature, making it hard to assess its significance and effectiveness. Q3: How does DRIFT compare to other autoformalization techniques in performance?
> >
> > We compare DRIFT to standard zero-shot [1] and retrieval-augmented baselines [2] used in the literature. **We utilize the dense passage retriever from RAuto [2] because it is the established state-of-the-art for dependency retrieval**, targeting the retrieval of specific formal objects required for formalization, rather than the similar theorems retrieved by standard k-NN or semantic baselines.
> > Regarding specialized systems (e.g., Goedel-Formalizer), we prioritized frontier models (GPT-4.1, DeepSeek-V3.1) because DRIFT is a reasoning pipeline **requiring advanced instruction-following capabilities often absent in smaller fine-tuned models**. Furthermore, our ConNF results demonstrate that generalization to unseen libraries requires the broader reasoning breadth of frontier models, whereas fine-tuned models are brittle outside their specific training distribution [3].
> >
> > *As requested, we will benchmark the finetuned Goedel-Formalizer-V2 (8B) and compare the performance. We update the results before the end of the rebuttal.*
> >
> > **Actions taken:** We evaluate Goedel-Formalizer and compare the performance with DRIFT. We will update the results before the end of the rebuttal.
> >
> > > Can the authors demonstrate gains on proof autoformalization using this method as well?
> >
> > We thank the reviewers for the suggestion, we think this is an interesting avenue for future research to explore. There is some evidence in related works that suggests that techniques employed in DRIFT may also transfer to proof search (e.g., ReProver [4], DeepSeek-Prover [5]). However, our work focuses on autoformalization.
> >
> > > Why was m=3 selected for the illustration stage? Is it possible that scaling this up to much more examples will improve more (e.g. in https://arxiv.org/abs/2404.11018)
> >
> > Thank you for the question. We selected $m=3$ to **optimize the trade-off between premise coverage and contextual noise**. In response to the effect of illustration budget, we conducted an empirical analysis of premise coverage rate versus illustrative theorem budget $m$ (see Figure 2, Appendix A.2.8). **As shown in Figure 2, m=3 represents the critical point of diminishing returns (the "elbow" of the curve)**. While coverage marginally increases up to m=5, the slope flattens significantly after m=3, indicating that the vast majority of discoverable premises are captured within the first three theorems. We deliberately stop at this efficient frontier because striving for the final ~5-10% of coverage requires adding more theorems, which disproportionately introduces irrelevant "distractor" premises and token overhead. We refer you to Section 5.3 (Table 3), where we observed that excessive context degrades performance in low-dependency regimes.
> >
> > **Actions taken:** We have added the empirical analysis of premise coverage rate versus theorem selection budget in Appendix A.2.8.
> >
> >
> > References:
> >
> > - [1] Zhang, Lan, Marco Valentino, and André Freitas. "MASA: LLM-Driven Multi-Agent Systems for Autoformalization." Proceedings of the 2025 Conference on Empirical Methods in Natural Language Processing: System Demonstrations. 2025.
> > - [2] Liu, Qi, et al. "Rethinking and improving autoformalization: towards a faithful metric and a dependency retrieval-based approach." The Thirteenth International Conference on Learning Representations. 2025.
> > - [3] Lampinen, Andrew K., et al. "On the generalization of language models from in-context learning and finetuning: a controlled study." arXiv preprint arXiv:2505.00661 (2025).
> > - [4] Yang, Kaiyu, et al. "Leandojo: Theorem proving with retrieval-augmented language models." Advances in Neural Information Processing Systems 36 (2023): 21573-21612.
> > - [5] Ren, Z. Z., et al. "Deepseek-prover-v2: Advancing formal mathematical reasoning via reinforcement learning for subgoal decomposition." arXiv preprint arXiv:2504.21801 (2025).

---

### Author Response · Authors · 2025-12-02
**Rebuttal Summary and Discussion (1/2)**

We thank the reviewers and AC’s for thoroughly evaluating our paper and rebuttal responses.
Below, we provide a summary of our responses to overlapping concerns across reviews, including clarifications on task scope, baseline selection and additional experimental results.

**Clarifications regarding the DRIFT pipeline and its contributions:**
- We would like to clarify that our primary contribution is the DRIFT pipeline itself: decomposing informal statements into granular sub-queries, leveraging the LLM's Lean knowledge to guide retrieval for each component, and providing illustrative theorems alongside retrieved premises. These design choices are novel and directly impact performance, as demonstrated in Table 3 where each component provides performance improvements. For retrieval, we follow the training recipe for the DPR model from Liu et al. (2025), which is an adjustable component in the framework.

**Additional experiments using finetuned open-source Formalizers:**
- We evaluated Goedel-Formalizer-V2-8B (Goedel-Formalizer) using its native instructions, instead of the prompt templates used for GPT-4.1 and DeepSeek-V3.1. For the Oracle setting, we adapted Goedel-Formalizer’s own template (using DRIFT’s prompts significantly reduced its performance in preliminary experiments).
- Goedel-Formalizer performs well on the in-domain MiniF2F-test benchmark (100% TC@10) but fails to generalize to the out-of-domain ConNF benchmark (only 10.93% BEq+@10). Moreover, it gains little from Oracle retrieval on ConNF (23.10% BEq+@10), unlike GPT‑4.1 and DeepSeek‑V3.1, which substantially benefit from including Oracle premises. This suggests limited instruction-following capacity for effective RAG use without further finetuning.
- We also tested Goedel-Formalizer as a decomposer. It largely ignored decomposition instructions, making it unsuitable in its current form. Using Goedel-Formalizer within DRIFT therefore requires (i) a different decomposition model or (ii) additional finetuning on DRIFT-style instructions. For context, GPT‑4.1 and DeepSeek‑V3.1 (with DRIFT) substantially outperform Goedel-Formalizer on the out-of-domain ConNF benchmark (e.g., 62.33% BEq+@10 vs. 10.93%) and show stronger single-pass correctness (pass@1) on ProofNet and ConNF.

| Dataset      | Formalizer              | Retrieval | TC@1  | BEq+@1 | TC@10  | BEq+@10 |
|--------------|-------------------------|-----------|-------|--------|--------|---------|
| ProofNet     | Goedel-Formalizer-V2-8B | Zero-Shot | 38.50 | 9.09   | 76.74  | 55.08   |
|              |                         | Oracle    | 35.29 | 9.09   | 75.40  | 62.03   |
| MiniF2F-test | Goedel-Formalizer-V2-8B | Zero-Shot | 93.33 | 22.22  | 100.00 | 93.33   |
|              |                         | Oracle    | 96.00 | 28.89  | 100.00 | 93.33   |
| ConNF        | Goedel-Formalizer-V2-8B | Zero-Shot | 16.03 | 2.29   | 71.19  | 10.93   |
|              |                         | Oracle    | 9.99  | 4.37   | 48.80  | 23.10   |

**Actions taken:** We have extended our analysis to include a comparison with the finetuned Goedel-Formalizer-V2-8B in Appendix A.2.10 (Table 10 and Figure 3, pp. 19--21).

**Clarifications regarding other retrieval methods:**
- Our task is dependency retrieval, not theorem retrieval. Tools like LeanSearch, Moogle, and BM25 usually identify semantically similar theorems, whereas DRIFT (and the RAuto DPR baseline we compare against) retrieves the **exact formal dependencies**, i.e. IsPrimitiveRoot, ZMod. Consequently, we strictly benchmark against RAuto (DPR), the established SOTA for this specific task. We provide additional details in the responses to the reviewers and Table R2.

**Ablation studies on demonstrative theorem budget and adaptive retrieval:**
- We calculated the premise coverage rate change as compared to the theorem top-m (see Figure 2 in Appendix A.2.8). As shown in Figure 2, m=3 represents the critical point of diminishing returns (the "elbow" of the curve). While coverage marginally increases up to m=5, the slope flattens significantly after m=3, indicating that the vast majority of discoverable premises are captured within the first three theorems. We motivate not adding more than m=3 demonstrative theorems in our ablation study (Table 3, Section 5.3), where we observed that excessive context degrades performance in low-dependency regimes. In response to the comment, we added the empirical analysis of premise coverage rate versus theorem selection budget in Appendix A.2.8.
- We address the potential of an adaptive retrieval gate by calculating the "Oracle Ensemble" performance, the theoretical upper bound of dynamically selecting between Zero-Shot and DRIFT. The ensemble consistently outperforms individual methods (e.g., raising Claude's BEq+@1 from 32.59% to 35.27%), confirming that the approaches are complementary. In response to the comment, we added the Potential for Adaptive Retrieval discussion in Appendix A.2.9.

---

> ### Author Response · Authors · 2025-12-02
> **Rebuttal Summary and Discussion (2/2)**
>
> **Qualitative analysis regarding scaffolding theorems and DRIFT surpassing Oracle:**
> - We added detailed qualitative analyses (Appendix A.3) confirming that DRIFT outperforms the Oracle on ConNF by providing syntactic scaffolding (e.g., correct namespace usage) via illustrative theorems (Appendix A.3.2). This also explains the observations in our ablation study (Appendix A.3.3). We also provided an error taxonomy for MiniF2F-test to explain why retrieval gains are smaller in self-contained problems (Appendix A.3.1).

---

### Meta-Review · Area_Chair_pkmC · 2026-01-06

**Summary:**

The paper proposes a four-stage framework for auto-formalization: statement decomposition, retrieval of formal dependencies, illustration of their usage, and completion of the formalization. All reviewers recognize the overall pipeline as novel, and agree that the illustration step leads to significant performance improvements.

The main concerns relate to the details of the decomposition module (Reviewers 1 and 2) and the lack of comparisons with related techniques (Reviewers 2, 3, and 4). Reviewer 4 gives the most negative rating, arguing that the method does not use frontier LLMs and newer versions of Lean, and that the evaluation metric (BEq+) may underestimate performance.

The rebuttal provides clarifications and new results that address most of these concerns. The authors include additional experimental results with Goedel-Formalizer-V2, address the lack of comparisons, and provide further details on the decomposer. They also justify the choice of Lean version, underlying LLMs, and the evaluation metric. These clarifications are likely to address Reviewer 4’s concerns.

Given the positive ratings from three reviewers and the high likelihood that Reviewer 4 will raise their score, I recommend acceptance.

**Reviewer Concerns:**

I believe the rebuttal addresses all major concerns.

**Reviewer Scores:**

Reviewers 2 and 3 have stated in the discussion that they will maintain their positive scores.

Reviewer 1 is very likely to maintain their score in line with Reviewers 2 and 3.

Reviewer 4 may raise the score, as concerns are addressed in the rebuttal.

---

### Decision · Program_Chairs · 2026-01-26

Accept (Poster)